# Factors in the Responsible Management of the Luna Valley Complex Geosite (NW Spain)—A Case Study

Esperanza Fernández-Martínez [1,*], Ismael Coronado [1], Luna Adrados [2] and Rodrigo Castaño [3]

1. Palaeontology Area, Faculty of Biological and Environmental Sciences, University of León, Campus de Vegazana s/n, 24071 León, Spain
2. GEOLAG, Turismo Geológico, Barrio La Vallina and 20-4ª, 33191 San Claudio, Spain
3. IGME-CSIC, Av. Real, 1, 24006 León, Spain
* Correspondence: e.fernandez@unileon.es

**Abstract:** The Luna Valley complex geosite (northwestern Spain) is a region of geoheritage significance located in an area with high environmental value. Geological studies began in the mid-20th century and continue to provide scientific data of significant relevance to the knowledge regarding the Palaeozoic stratigraphy of northern Gondwana and the tectonics of the Variscan orogen. This region also has high value for geoeducation, being visited regularly by both students and the general public. Educational use of the area has promoted the creation of several publicly available materials and activities that include trails, guides, displays and brochures, as well as the development of a small museum. However, over time, weathering; the abandonment of rural life; and the intensive, uncontrolled, and careless use of this region as a geosite for scientific and educational purposes has led to significant degradation and the consequent loss of its geoheritage value. This paper describes the geology of five key geosites in the Luna Valley. This is followed by a review of the promotional initiatives carried out in the area. These data, along with our knowledge of the area, allow us to develop a heritage analysis that includes the main geological interests, conservation status and some key management issues for each of these five individual sites. Several recommendations aim to control the physical degradation of the geosites, encourage their regular monitoring and the updating of the outreach materials using virtual tools, and promote the involvement of the local population in the conservation of this unique site.

**Keywords:** anthropic vulnerability; geoheritage; Palaeozoic; protected areas; risk of degradation

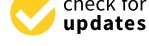



## 1. Introduction

Geoheritage refers to the elements and features of the Earth that are considered to have significant value [1]. Although an awareness of the need to protect this heritage dates back to the beginnings of nature conservation, the Digne Declaration (Declaration of the Rights of the Memory of the Earth) was a major milestone in the recognition of Earth heritages [2]. From that moment on, different working groups have been developing concepts and methods related to geological heritage, with very different achievements, in several countries [3]. Another important milestone is the recognition of geological heritage and geodiversity by the International Union for Conservation of Nature (IUCN), by creating, in 2013, a Geoheritage Specialist Group within the work of WCPA (Word Commission on Protected Areas). However, and despite the work done, compared with biodiversity, the conservation and management of geoheritage has only very recently begun to be considered in a more structured way. The establishment of the UNESCO Global Geoparks in November 2015, with 195 Member States, is evidence of the governmental recognition of the importance of managing outstanding geological sites (geosites) and landscapes in a holistic manner [1,4–6]. Geosites are points, sections or areas containing important geological features that usually have been selected through an audit, selection

and assessment process [1,4–9]. Geosites features can be locally, nationally or internationally important, and they could have purely scientific (=intrinsic) value or have educational, recreational, spiritual, aesthetic, or other value [10,11]. Many geosites can be used as a basis for geotourism development even in remote regions [12,13]. Geosites need some exposure to be included as geological heritages, and many of them include dynamic processes [1,3]. Despite the apparent robustness of the Earth's geology, many geological features are subject to both natural threats and human interventions [9,14,15]. Therefore, the management of these sites, i.e., geosites of geoheritage significance, requires the development of specific conservation plans [16,17]. In this context, the 2015 and 2020 editions of the guidelines for geoconservation in protected and conserved areas [1,18] are landmarks that facilitate the design of geoconservation actions. Within these actions, monitoring is the key to assessing progress in any conservation plan [15,16,19].

In Spain, geoheritage was introduced for the first time in the natural laws in 2007 (Ley 42/2007 de patrimonio natural y de la biodiversidad). Driven by this law, a national and several regional inventories were developed, and some geoconservation strategies started to be implemented. The national inventory, called IELIG (Inventario Español de Lugares de Interés Geológico), has a strong conceptual basis and a very elaborate methodology [20]. However, there are still no comprehensive management plans for protected areas that include geological heritage, and nor have general geoconservation strategies been developed, as is the case in other countries such as the UK [16].

The focus of this work is on the conservation and public use status of those sites in the Luna Valley that, despite being in protected areas, lack management and geoconservation plans. To approach this goal, we analyse the case of five outcrops of Precambrian and Palaeozoic rocks located in the middle sector of the Luna River Valley. These geosites have been used for more than 50 years for scientific and educational purposes and have been enhanced for recreational activities, but they still lack a management plan and are not included in the conservation plans of the protected areas to which they belong. The results obtained from this study allow us to make some management recommendations for this geosite. Moreover, as the issues at this site are very similar to those at many other geosites, these suggestions may be generally applied in other areas.

## 2. Description of the Case Study

As mentioned earlier, this work focuses on a set of outcrops of Precambrian and Palaeozoic rocks located in the middle sector of the Luna Valley (southern slope of the Cantabrian Mountain range, northwestern Spain, Figure 1) within the municipality of Los Barrios de Luna (province of León, autonomous region of Castilla y León, Spain). This area can be regarded as a complex area [7] or complex geosite [21] that includes several single geological features that have been the subject of numerous scientific works [22–42]. As a result of this research, this region has become a classic reference area for the study of the Palaeozoic in Iberia [43].

The geological features included in this complex area show a diverse management typology that includes scenic viewpoints, points of interest, type sections and simple areas (for an explanation of these different management types or categories, see Fuertes-Gutiérrez and Fernández-Martínez, 2010) [7]. Five single geosites are of particular interest (Figure 2) for both scientific and geoeducational purposes and will be described and analysed in this work:

Geosites 1 (a viewpoint) and 2 (a point) show an angular unconformity between Precambrian and Cambrian rock sequences.

Geosite 3 is a fairly complete section of a Lower Palaeozoic (Cambrian to Silurian) sedimentary rock outcropping on the slope of a local road.

Geosite 4 is the continuation of geosite 3 with Silurian to Middle Devonian rocks on the slope of a regional road.

Geosite 5 is an area where a well-known site of middle Cambrian trilobites and echinoderms is located.

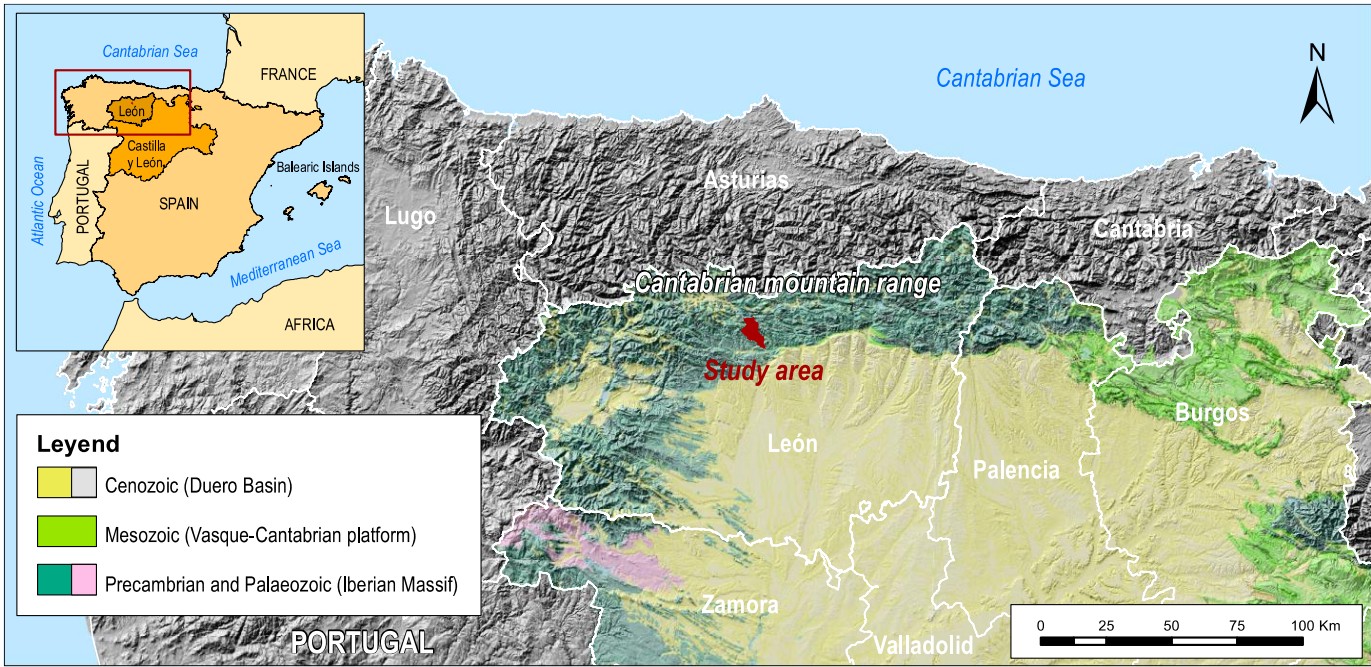

**Figure 1.** Geographic setting of the Luna Valley. Top left: map of the Iberian Peninsula indicating the autonomous region of Castilla y León and the province of León. Background: digital elevation model showing the northwestern Iberian Peninsula with the delimitation of several provinces. A general geology of the autonomous region is also shown. Note the extension of the Cantabrian Mountain range and the situation of the studied area on the southern slope of this range. Source: prepared by the authors. MDT: https://booksite.elsevier.com/9780444534477/DCW_Europe_WGS84.php (accessed on 16 November 2022); Geological background: http://idecyl.jcyl.es/ (accessed on 16 November 2022).

This complex geosite is listed in the Global Geosite catalogue as PZ002, within the geological framework "Lower and Middle Palaeozoic stratigraphic successions of the Iberian Massif" [44,45]. They also appear with the code CA002, "Precambrian and Palaeozoic rocks of the Luna Valley" in the Spanish Inventory of Geosites (Inventario Español de Lugares de Interés Geológico, IELIG) http://info.igme.es/ielig/ (accessed on 16 November 2022). (Figure 2). In addition, they are included in the provincial inventory of León [46] and in a local inventory of geological heritage [47].

Four of the five individual geosites are located in the Babia y Luna Natural Park, and all of them belong to the Biosphere Reserve of Los Valles de Omaña y Luna (Figure 2).

Due to its location in protected natural areas, and according to the Spanish laws, part of its management is carried out by the administration of the autonomous region of Castilla y León. However, their recognition and dissemination have mainly been carried out by the Cuatro Valles Association (https://www.cuatrovalles.es/), a nonprofit organisation made up of public and private operators whose objective is to develop its member municipalities, all of which have a markedly rural character.

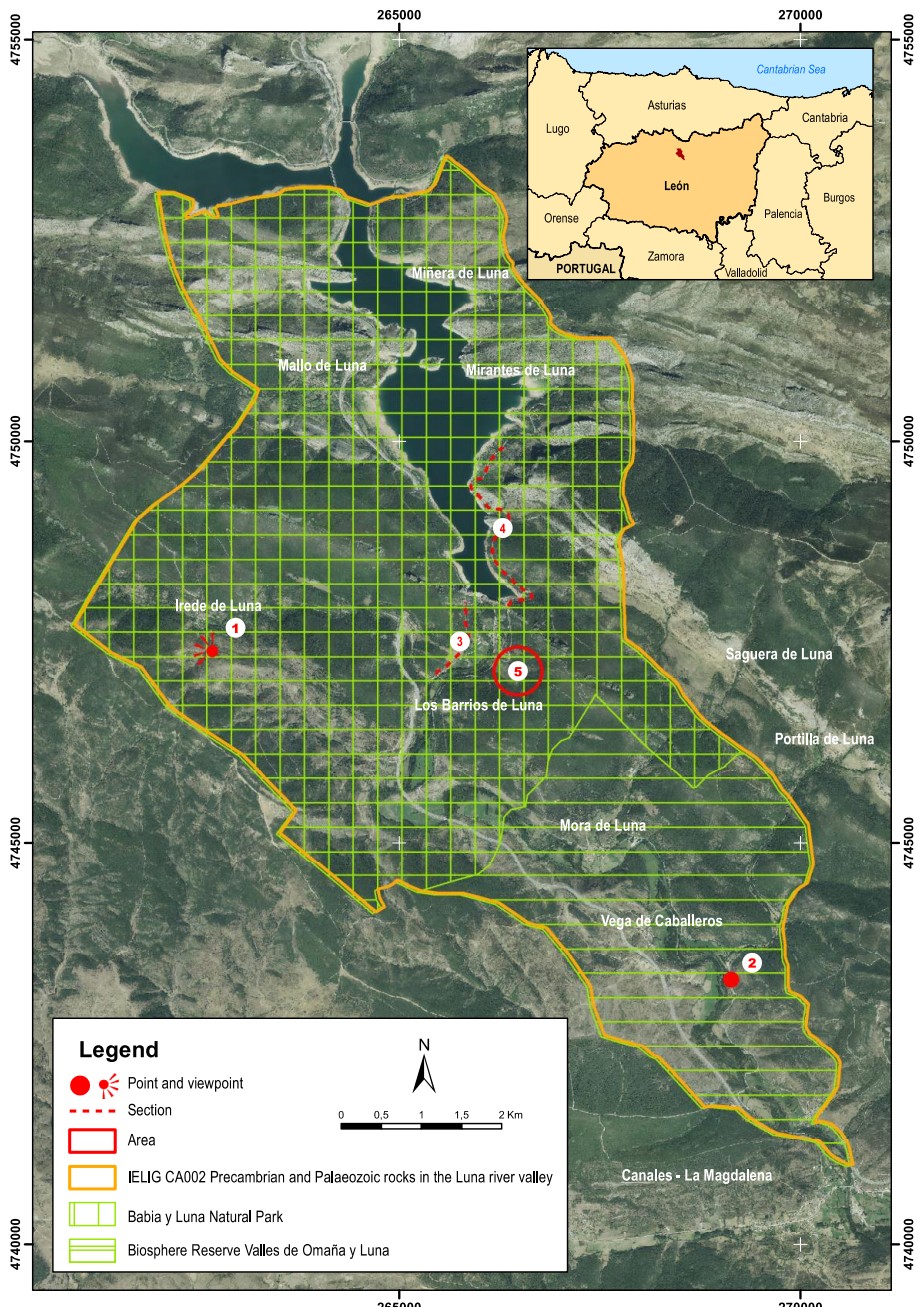

**Figure 2.** Heritage setting of the complex geosite Precambrian and Palaeozoic rocks of the Luna Valley. Top right box: geographic location of the geosite in the province of León, northwestern Spain. Geosite delimitation according to the Spanish inventory of geosites (IELIG, CA002). Some reference localities have been included with their names in white. The areas of the geosite belonging to the Babia y Luna Natural Park and the Valles de Omaña y Luna Biosphere Reserve have been superimposed on this delimitation and marked. Red numbers indicate the five single geosites analysed in this paper: 1. Viewpoint showing the angular unconformity between Neoproterozoic and Cambrian rocks at Irede de Luna. 2. Angular unconformity between Neoproterozoic and Cambrian rocks near Portilla de Luna. 3. Lower Palaeozoic stratigraphic section. 4. Middle Palaeozoic stratigraphic section. 5. Middle Cambrian palaeontological site. Typology of these geosites (point and viewpoint; section and area) after Fuertes-Gutiérrez and Fernández-Martínez [7]. Source: prepared by the authors. Orthophoto: http://orto.wms.itacyl.es/WMS (accessed on 16 November 2022).; PNS: https://idecyl.jcyl.es/geoserver/ps/wms (accessed on 16 November 2022).

From a geological point of view, this area began to be studied in the 1950s by geologists from the University of Leiden, with the first papers published in the 1960s [22–24,34,36–42]. Some synthesis of the investigations of the first decades can be found in Aramburu et al. [25–27]. Of particular relevance are data on Palaeozoic stratigraphy and palaeontology in northern Gondwana [28–30]. In addition, there are several palaeontological sites in the area that have yielded diverse palaeobiological, palaeoecological and chronostratigraphic data about the Palaeozoic marine biota [31–33,35].

The scientific, educational and touristic value of this complex geosite, as well as its inclusion in heritage inventories, have provided several publicly available materials (mainly guides, panels, videos, and a museum) and various onsite activities related to the geology of the area. The different authorities involved have carried out these dissemination materials and actions independently. The lack of coordination between them and the lack of a comprehensive and well-defined management plan for this complex geosite are remarkable. There are also no studies that analyse the heritage aspects of this geosite, such as its state of knowledge and use, the results of its dissemination and the natural and anthropic degradation processes that have an impact on it.

At present, the five single geosites mentioned above (Figure 2) show significant degradation in processes, with a consequent loss of value. In the case of the lower Palaeozoic section (Geosite 3, Figure 2), and although not exclusively, much of its degradation seems to be linked to its educational and touristic use as a heritage element; the other outcrops are at risk of degradation due to natural and anthropic causes unrelated to their scientific, educational or tourist uses.

The aim of this work is to highlight, discuss, and provide some key points on the need for conservation in light of the adverse environmental damage due to public use in a region with geoheritage significance but without appropriate management strategies in place.

## 3. Methods

The method employed in this work can be summarised in the following steps:

1. Compilation of up-to-date geological information (Figures 3–5).
2. Compilation of data on the educational and recreational use of each single geosite. For this purpose, it was essential to carry out a historical search of the administrations and companies that have worked in the area. Subsequently, they were contacted to compile information on the activities carried out and on the printed and online publications.
3. Photo monitoring. Evaluation of the state of conservation using photographs of certain geological features taken between 2008 and 2022 and comparing the conservation status of these elements. The monitored features are the main geological elements of the simple geosites (for instance, an oncoid bed or a surface with ripples). The main aspects monitored were the growth of vegetation on slopes, lichen and moss growth on the rock itself, the occurrence of cracks and rock-falls, hammer marks, the disappearance of geological features and the presence of paint or marker marks on the rock.
4. Review of the regulations of the protected area where the geosite is located. Compilation of proposals for action and meeting with the managers and some stakeholders of this territory to study the possibilities of implementing these proposals.

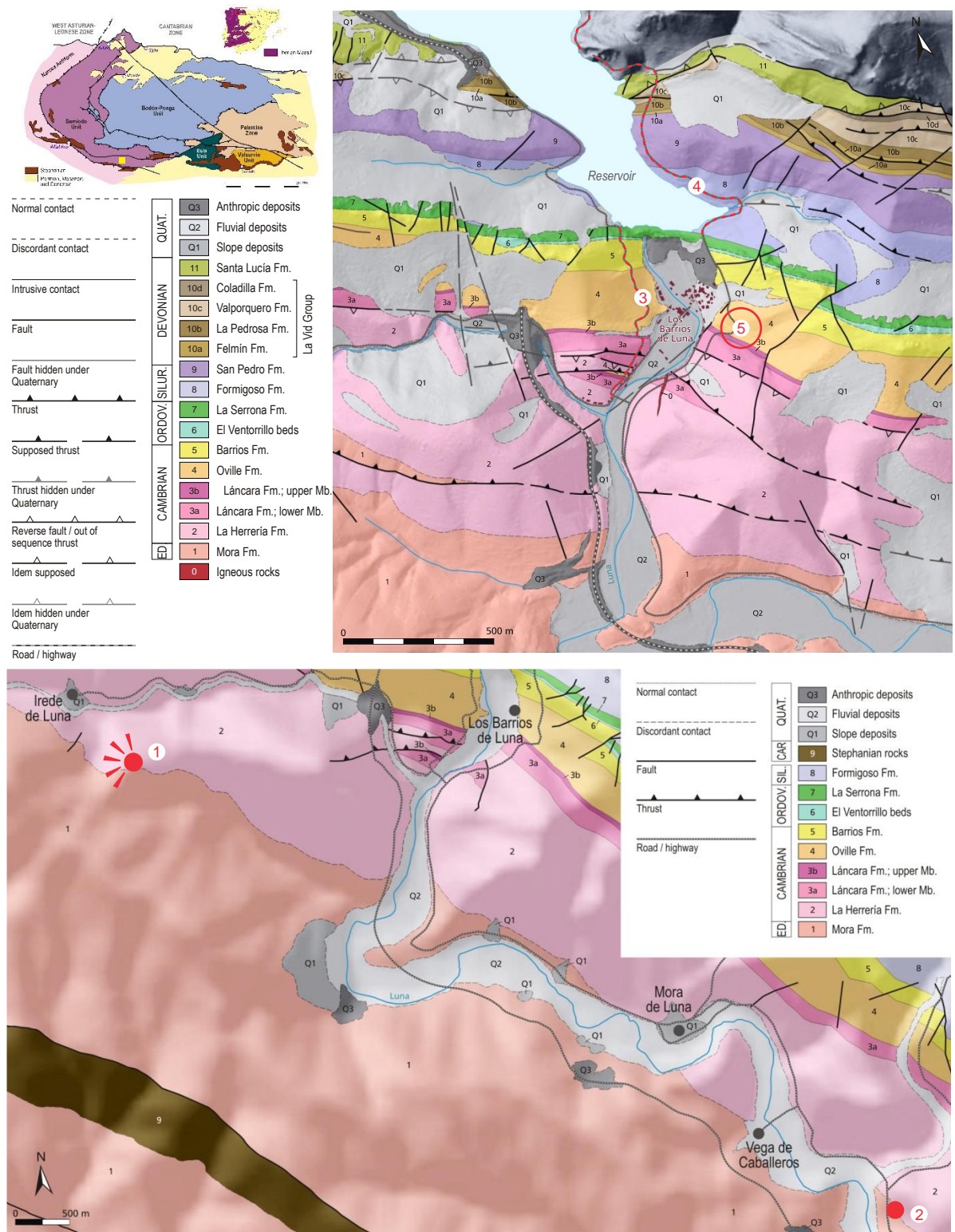

**Figure 3.** Geological location and maps of the complex geosite Luna Valley with indications of the location and typology of each of the five single geosites (same legend as Figure 2). Source: prepared by the authors.

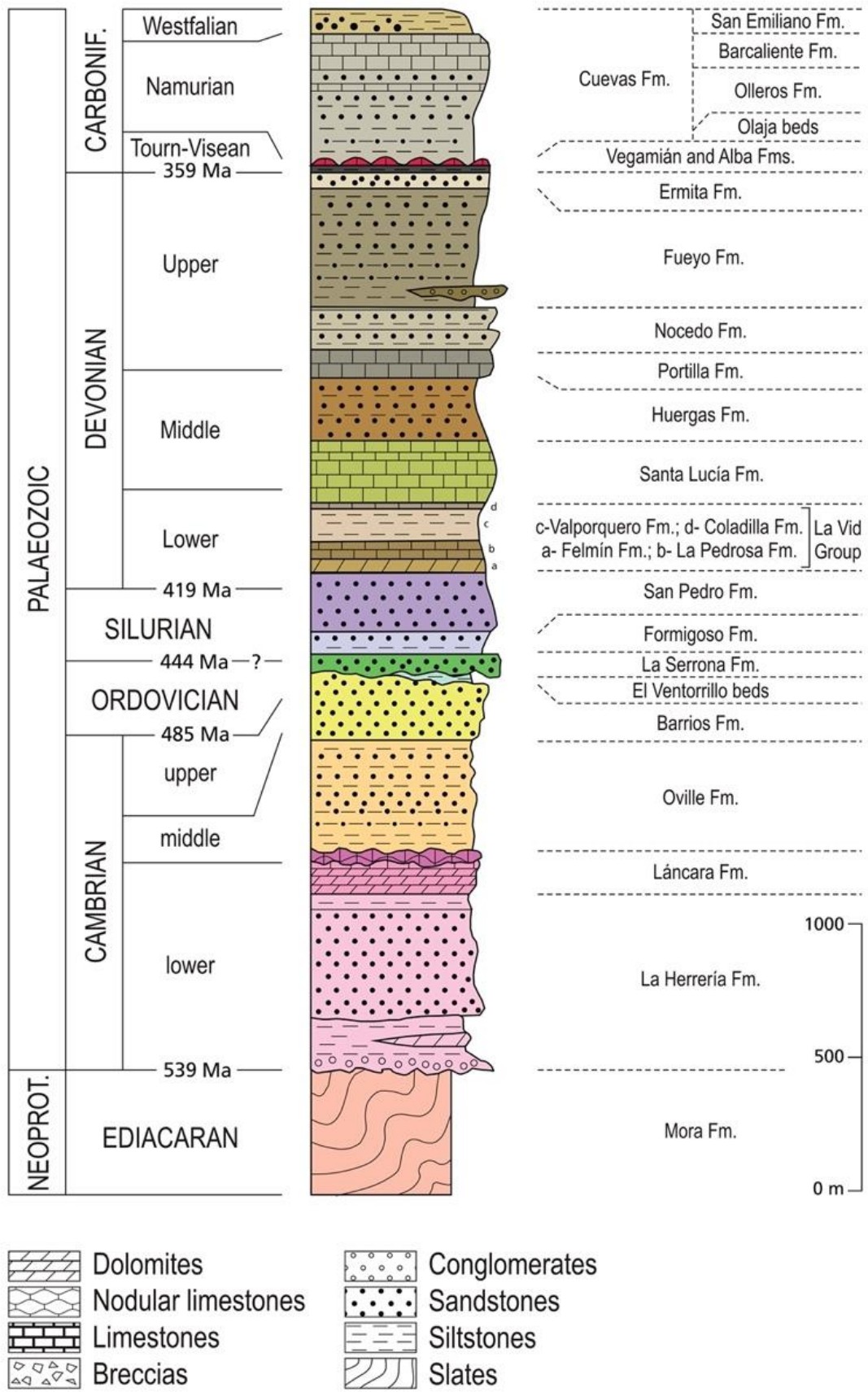

**Figure 4.** Stratigraphic column of the Palaeozoic rocks of the Cantabrian Zone with the lithostratigraphic units outcropping in the Luna Valley complex geosite. Source: prepared by the authors.

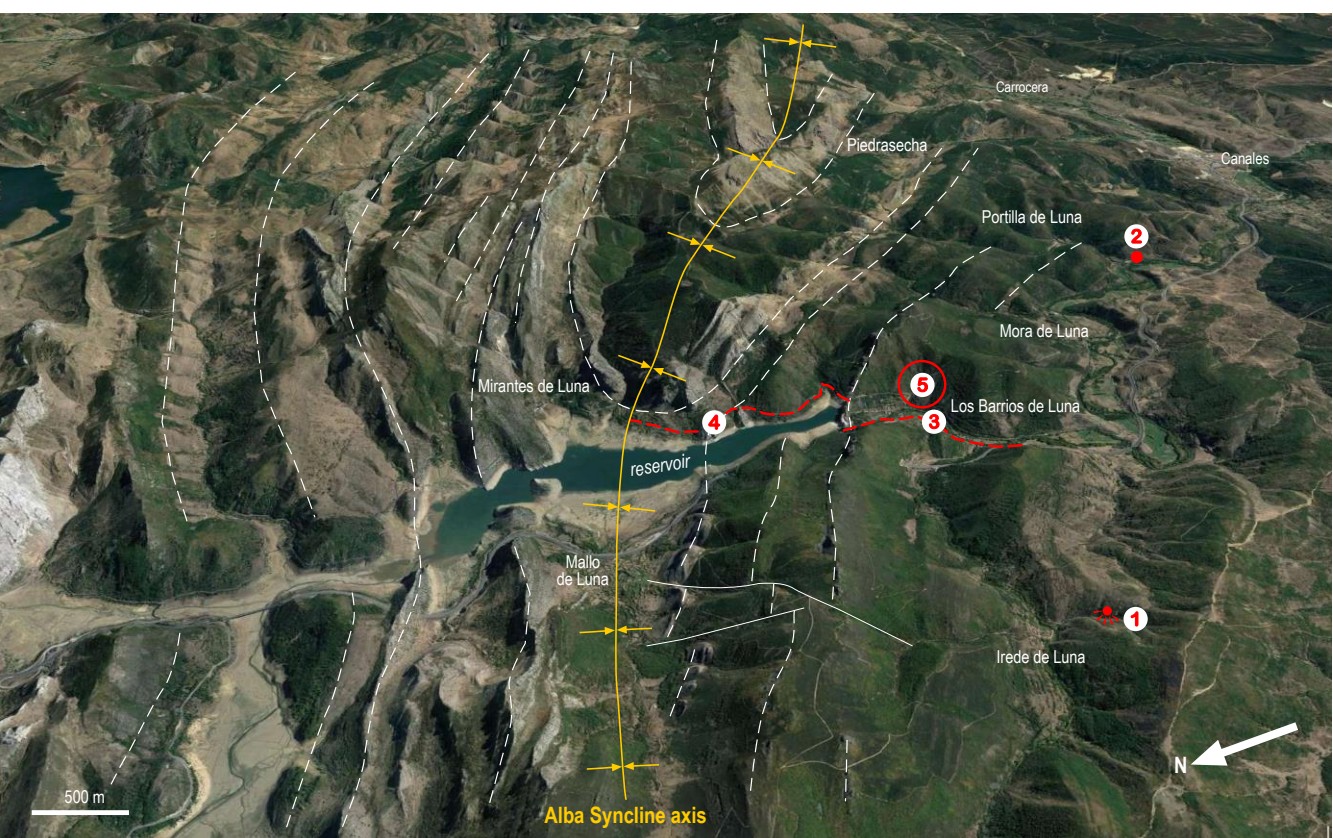

**Figure 5.** Orthophoto showing the Alba syncline structure and the location of the five single geosites described in this study. 1. Viewpoint at Irede de Luna. 2. Point near Portilla de Luna. 3. Lower Palaeozoic stratigraphic section. 4. Middle Palaeozoic stratigraphic section. 5. Middle Cambrian palaeontological site. Source: prepared by the authors. Template of the map: Google Earth.

## 4. Results

### 4.1. Geological Setting of the Study Area

The complex geological site of Luna Valley, located in the Iberian Massif, is a large outcrop of Precambrian and Palaeozoic rocks that run through the west of the Iberian Peninsula and form the basement of Iberia (Figure 3). Placed in the north of the Iberian Massif, the Cantabrian Zone is a foreland thrust-and-fold belt of the late Palaeozoic Variscan collisional orogeny. This deformation took place under shallow crustal conditions and is made up of mainly sedimentary rocks; its structure shows a thin-skinned geometry, with more than one level of decollement and a wide range of different thrust systems [22]. However, the creation of the present relief of the Cantabrian Mountains is mainly due to the tectonic uplift of the ancient Variscan basement in favour of a large northward-sloping thrust, the age of which is not known but is about 30 Ma [48].

The middle course of the Luna River is located in the extreme southwest of the Cantabrian Zone (Figure 3), more precisely in the Somiedo Unit [23]. The oldest rocks in this area (Mora Formation) are alternating slates and sandstones, Neoproterozoic in age, that have been affected by low-grade metamorphism. An angular unconformity brings the Precambrian materials in contact with the overlying Palaeozoic rocks. The Palaeozoic series consists of more than 5000 m of sedimentary rocks, including carbonate rocks (limestones, dolomites), siliceous rocks (mainly shales and sandstones) and mixed carbonate/siliceous rocks (marlstones). The age of this series ranges from Cambrian to Middle Devonian, although some early Carboniferous rocks are also found in the area. Cambrian to lower Carboniferous rocks are marine sediments deposited on the extensive northern margins of Gondwana. To date, the Palaeozoic rock outcroppings in this area have been assigned to 24 formations and 2 informal units (Figure 4).

Source: prepared by the authors, map template GEODE, Mapa geológico digital continuo de España (online https://info.igme.es/cartografiadigital/geologica/Geode.aspx) (accessed on 29 September 2022).

Tectonically, the Somiedo Unit is made up of an imbricated system of thrusts that define a synformal structure. The geosites discussed in this work are located on the south flank of a minor synformal structure known as the Alba syncline (Figure 5). Both flanks of this syncline are affected by numerous fractures, and minor tectonic sheets have been identified.

From a geomorphological point of view, the Luna Valley shows a typical Appalachian relief, with intercalations of carbonate and siliciclastic rocks and the presence of more and less erodible rocks. These alternations generate a mixture of colours and textures of high aesthetic value. Limestones usually form the main relief, while the more easily eroded siliciclastic rocks (mainly sandstones and shales) give rise to more fertile areas occupying the main valleys and hills (Figure 5). In the landscape, the quartzite crest of the La Serrona Formation stands out: it has been used to close the Los Barrios de Luna reservoir.

There is no evidence of glaciers in the area, with erosion by the river before its damming being the main geological agent. The limestones show some development of karst, but it is not very marked. Weathering occurs mainly due to sharp temperature contrasts, root action and slope dynamics (i.e., gravitational processes).

### 4.1.1. Geosites 1 and 2. Angular Unconformity between Neoproterozoic and Cambrian Rocks

The oldest rock outcroppings in the complex geosite of Luna Valley are of the Neoproterozoic age [49] and correspond to the Mora Formation (Figure 4), which stratotype is located in Mora de Luna, near single Geosite 2. It is made up of alternating sandstones and slates with a very low degree of metamorphism. These rocks are usually deformed by asymmetric folding at different scales and show a marked associated cleavage (Figures 6–8). The upper lithostratigraphic unit is the La Herrería Formation (Figure 4), whose base is Ovetian (Cambrian, epoch 2, age 3, ca. 521 Ma); it consists of sedimentary rocks, mainly sandstones with some interbedded shales. The contact between the Mora Formation and the La Herrería Formation is an angular unconformity that can be seen mainly at geosites 1 and 2.

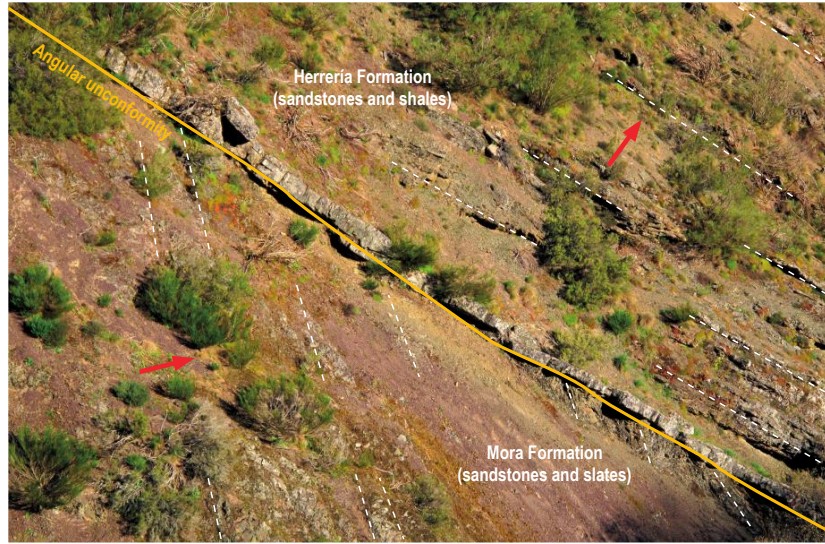

**Figure 6.** Angular unconformity between the Neoproterozoic–Cambrian rocks from a viewing point near the town of Irede de Luna. The oldest rocks are grouped in the Mora Formation and consist of sandstones and slates with little metamorphism. The Cambrian rocks belong to the La Herrería Formation, whose base is made up of sandstones with some interbedded shales. The yellow line indicates the angular unconformity. White dashed lines show the bedding planes. Source: prepared by the authors.

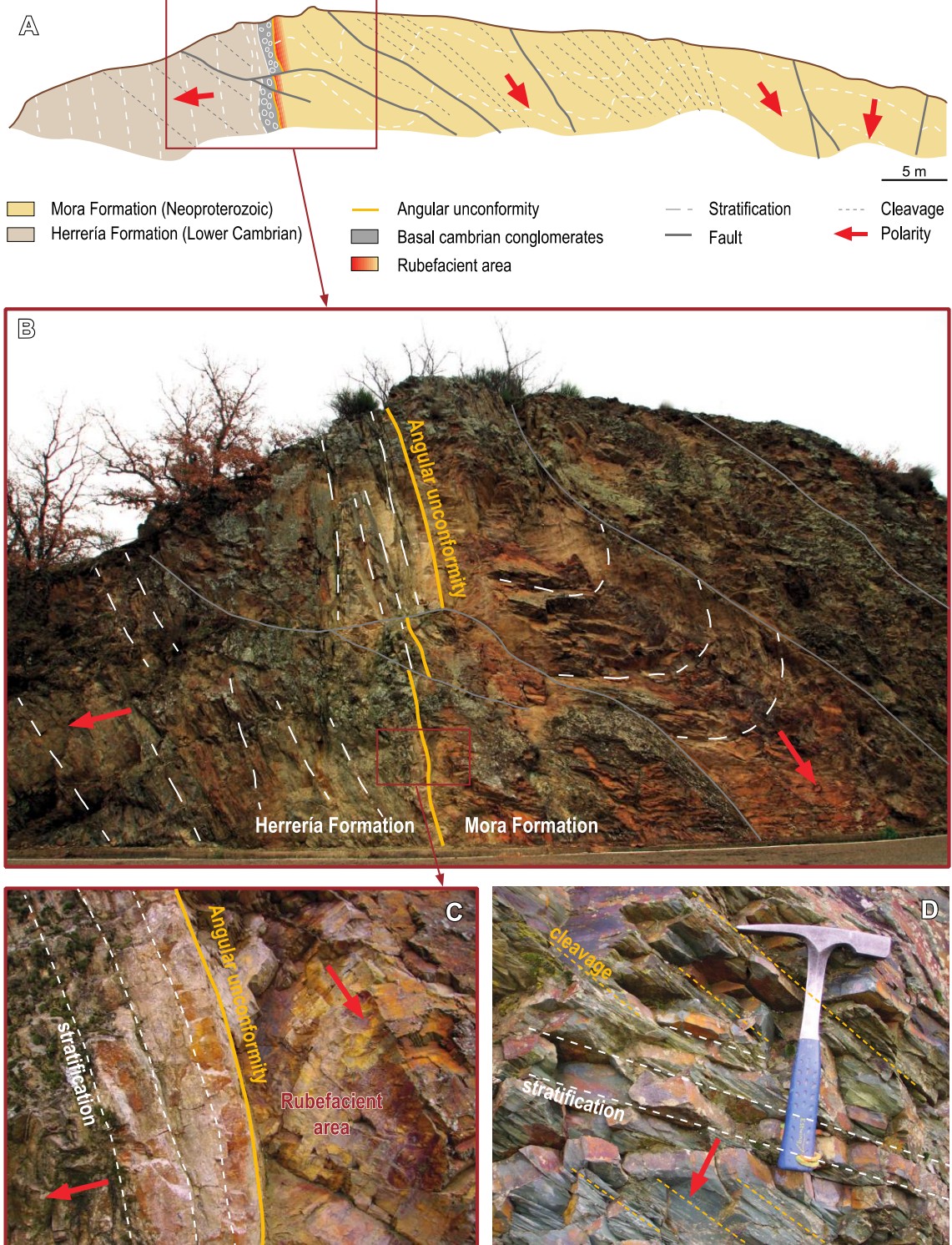

**Figure 7.** Angular unconformity between the Neoproterozoic–Cambrian rocks from a point near the town of Portilla de Luna. The yellow line indicates the angular unconformity plane. White dashed lines show the bedding planes. Dark lines show some faults. Red arrows indicate the position of the lower boundaries of the strata (stratigraphic polarity). (**Bottom left**): Detail of the unconformity plane shown in picture above. (**Bottom right**): Detail of the lithology of the Mora Formation; note the greenish slate and the occurrence of some cleavage in these rocks. Source: prepared by the authors.

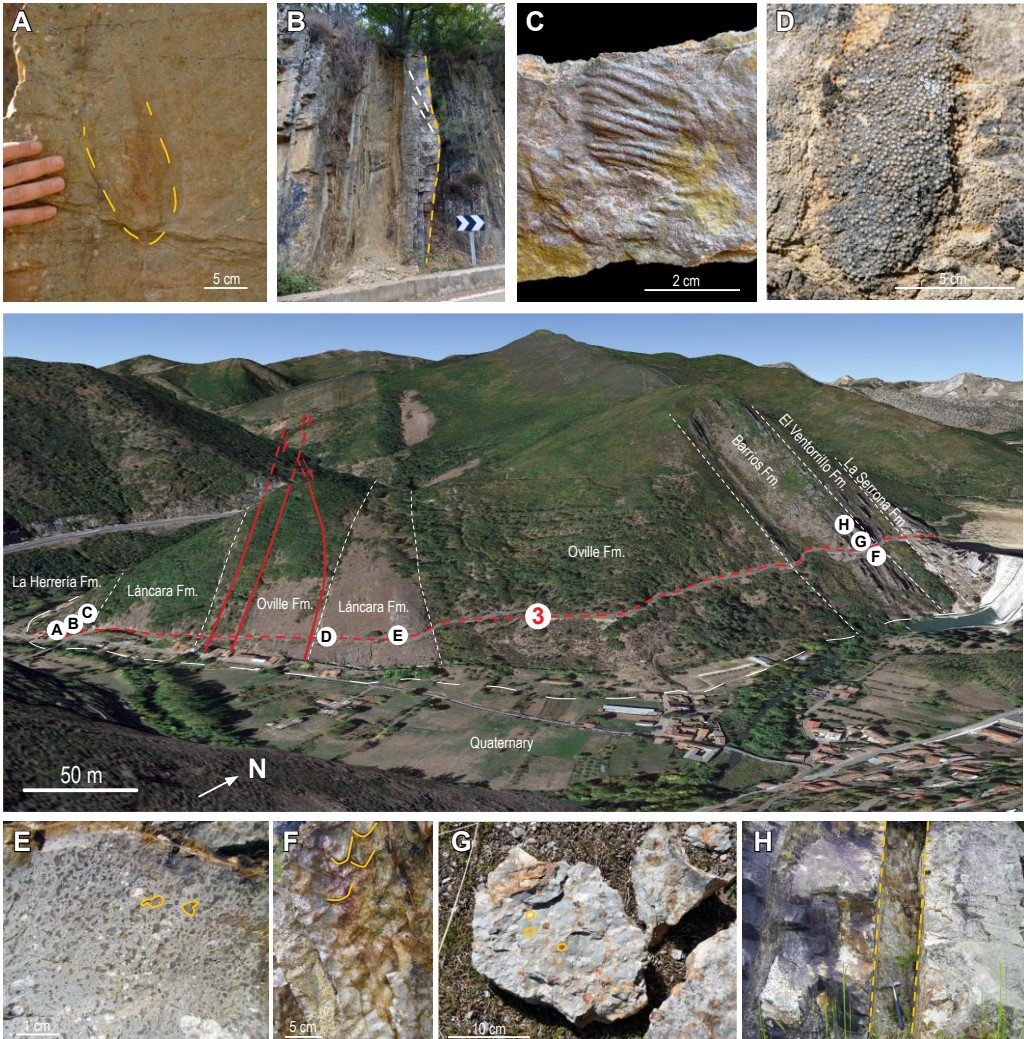

**Figure 8.** Orthophoto showing Geosite 3, the classic lower Palaeozoic section at Los Barrios de Luna, with the different outcroppings. A-H, some geological features of interest and their distribution in the section. (**A**) Current crescent. (**B**) Megaripple. (**C**) *Monomorphicnus* isp. (**D**) Microbialites type oncolites. (**E**) Fenestral porosity (type bird eyes). (**F**) Ripples (photo taken in 2009). (**G**) *Skolithos* isp. (**H**) Tonstein. Source: prepared by the authors. Template of the 3D image: Google Earth.

**Geosite 1** is a classic site located near the graveyard of the town of Irede de Luna (Figures 2–6); it allows direct viewing of the angular unconformity between the two formations. In this outcrop, the strata of the Mora Formation dip about 60° to the NE, and above them, the sandstones and shales beds of the La Herrería Formation form an angle of about 20° (Figure 6).

**Geosite 2** (Figures 2–5 and 7) is a point located on the slope of the C-623 road in the vicinity of Portilla de Luna. Here, the unconformity is vertical, with the Mora Formation layers showing intense folding in strong contrast to the subvertical La Herrería Formation layering (Figure 7). In addition to showing the tectonic relationships between the two formations, this geosite illustrates particularly well the oldest rocks of the Cantabrian Zone and their low-grade metamorphism. It also illustrates their tectonic structures, characterised by asymmetric folds (at different scales), many faults and marked axial-plane cleavage. The lithological rhythmicity observed in the Mora Formation and the occurrence of several sedimentary structures including flute casts, bounces and slumps suggest a deposit formed by turbidity currents.

The main tectonic elements in the Mora Formation (folding, cleavage), its low degree of metamorphism and the angular unconformity itself have their origin in the Cadomian orogeny, which occurred at the end of the Neoproterozoic Era.

At Geosite 2, the upper part of the Mora Formation shows a rubefacient process (reddening of rocks) whose origin is still unclear. Since it only affects the rocks of the Mora Formation, it has been postulated that it could be a case of subaerial alteration due to the emersion of these rocks at the end of the Precambrian (Cadomian orogeny). However, this colour has also been interpreted as the possible result of alteration by fluid circulation during later tectonic events.

4.1.2. Geosites 3 and 4. Classic Stratigraphic Section around Los Barrios de Luna

These two geosites constitute a single section, but it is typical to divide them into single geosites because of their different heritage value. The first part is more complete and is of much greater scientific, educational and touristic interest than the second part of the section.

Aramburu et al. [50] provides a good stratigraphic description of the lower Palaeozoic succession in the Cantabrian Zone, while Castaño de Luis and Toyos [51] provide an updated summary of this section.

Geosite 3 (Figures 2–5, 8 and 9) is 1.5 km long and contains rocks from the early Cambrian to (probably) the Late Ordovician, according to the following outline:

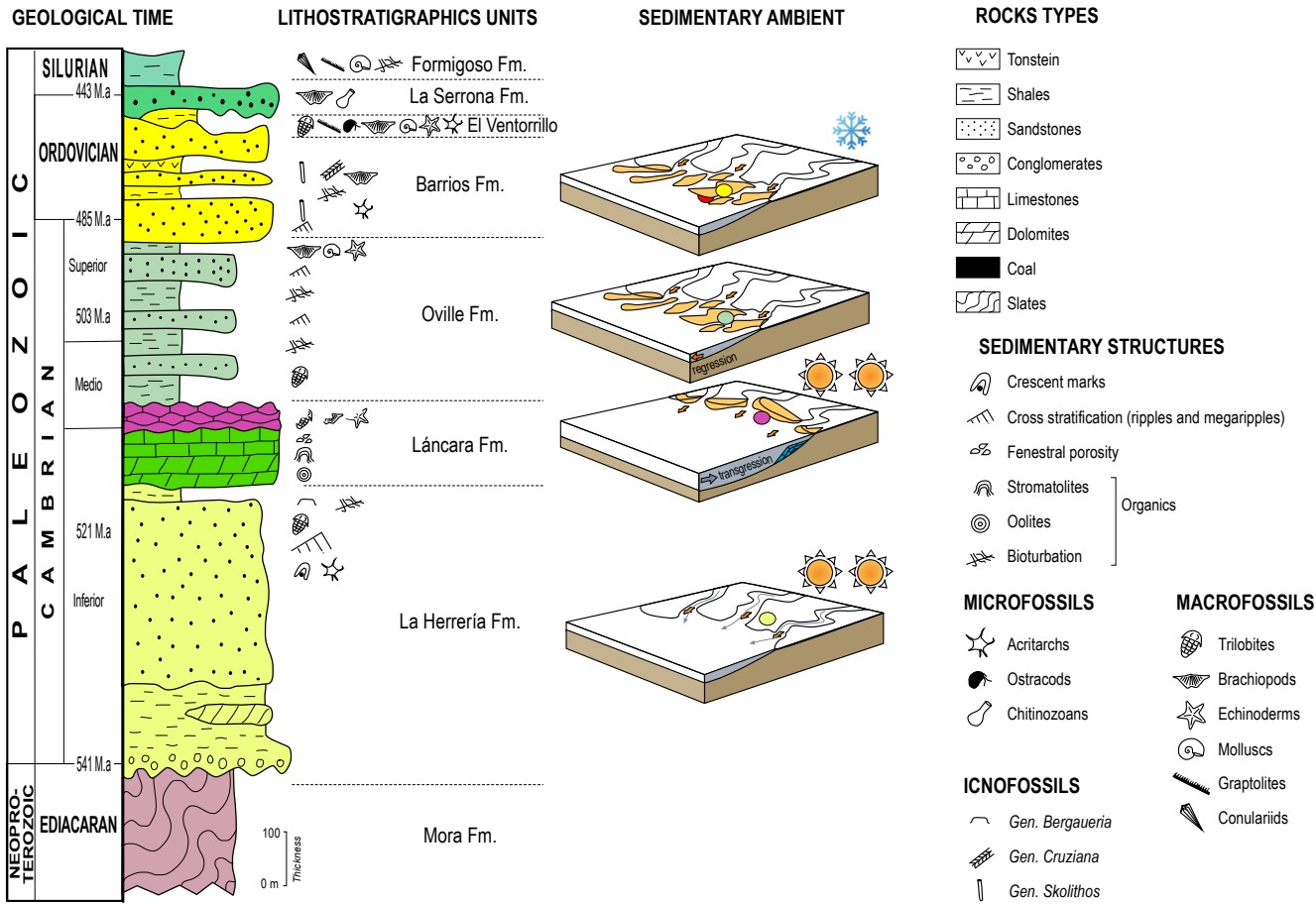

**Figure 9.** Stratigraphic column used in teaching activities in Geosite 3 showing the lithostratigraphic units and the key geological features (stratigraphic structures and fossils) that appear in these rocks. These features, together with the lithological variations, are used to interpret the environmental changes recorded in these rocks. Source: prepared by the authors.

1.  Upper member of the La Herrería Formation (named Barrios Beds because of the excellent outcrop in this section). It consists of alternating shale, sandstone and some dolomite beds. Some sedimentary structures such as current crescent, cross and parallel stratification, ripple surfaces and some good examples of megaripples also occur. Some layers are very rich in ichnofossils, including *Astropolichnus hispanicus*, a soft-bodied cnidarian cubicnus that characterises the Ovetian stage (lower Cambrian). These layers are regarded as having been deposited in a shallow marine environment.

2.  Láncara Formation. A set of carbonate rocks, with a lower member formed by dolomites and limestones and an upper member consisting of a characteristic red limestone (i.e., griotte limestones). The most common sedimentary structures are microbialites and other carbonate rocks (algal laminations, stromatolites, oolites, etc.) and karstic cavities of different sizes filled with calcite. The red limestones of the upper member are rich in fossils of benthic organisms, mainly trilobites, brachiopods, echinoderms and sponges. Some of these fossils provide dates for these rocks (viz., middle Cambrian). These materials were deposited in a deepening sublittoral marine environment and with the red limestone representing a condensed series.

3.  Oville Formation. Consists mainly of alternating fine-grained sandstones and shales, with layers of white quartz sandstones towards the top. Ripples and layers with abundant bioturbation are very frequent. The lower member contains trilobites and echinoderms, but in this section, the beds are not especially fossiliferous, and for this reason, the study of this fauna is carried out in Geosite 5. In the middle member, ichnofossils are very common, while the upper member has yielded some acritarchs. The latter have dated the top of the formation as Languedocian age (middle Cambrian). The Oville Formation records a process of shallowing from a sublittoral marine environment to a braided plain delta.

4.  Barrios Formation (its stratotype is in this section, although the upper part is partially covered by debris). Consists of white quartz sandstones (cemented by quartz) and intercalated grey or greenish shale. The most visible sedimentary structures are parallel and feature cross lamination, cross stratification and both wave and current ripples. In the upper part, there is an interval of about 5 m with abundant bioturbation, including some beds with the ichnofossil *Skolithos*. Above this interval, a 30 cm tonstein of kaolinite occurs. This formation is poor in macrofossils, but some shaly levels have yielded acritarchs. Studies of these acritarchs [27] and the ichnofossils suggest a middle-late Cambrian age for the base of the formation. However, the kaolin layer, located near the top, has been radiometrically dated at 477 Ma as Early Ordovician (Tremadocian–Floian boundary [52,53]. From a sedimentological point of view, this formation is interpreted as the coastal and alluvial deposits of a braided plain delta.

5.  El Ventorrillo beds. Characterised by dark shales and fine-grained sandstones with ferruginous crusts. They are very rich in both macrofossils (trilobites, graptolites, brachiopods, echinoderms and molluscs) and microfossils (ostracods, acritarchs and chitinozoans). These fossils date the layers as Middle Ordovician.

6.  La Serrona Formation. Made up of quartzite with some red shales and abundant pyrite nodules. Their resistance to erosion produces very sharp relief. No fossils have been found in this section, and the sedimentary structures typical of coastal environments only appear in the upper part of the formation. The quartzites have been interpreted as fillings of palaeovalleys generated in the underlying rocks during the lowering of the sea linked to the end-Ordovician glaciation.

Both El Ventorrillo beds and La Serrona Formation were defined by Toyos and Aramburu [29], and their stratotypes are located in this section.

The stratigraphic information recorded in this section, together with other nearby sections, made it possible to interpret the environmental changes that took place during the deposition (Figure 9). This section continues upstream from the closure of the dam, on the CL-626. This part of the section (Geosite 4 in this paper, Figures 2–5 and 10) is

approximately 1.2 km long and displays Silurian and Devonian rocks, organised into 6 lithostratigraphic units, all of them characteristic of the Cantabrian Zone: Formigoso Formation, San Pedro Formation, La Vid Group, Santa Lucía Formation, Huergas Formation and Portilla Formation.

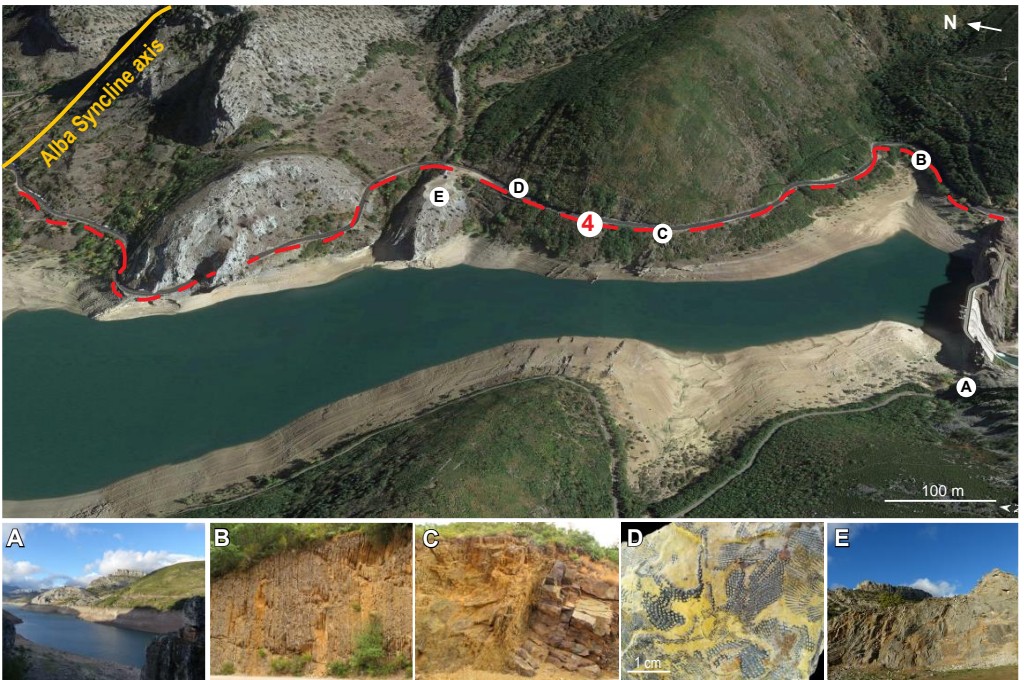

**Figure 10.** Orthophoto showing Geosite 4, a middle Palaeozoic section near Los Barrios de Luna and location of some geological features. (**A**) View of the Geosite 4 from Geosite 3. (**B**) Outcrop of the Formigoso Formation. (**C**) Outcrop of the San Pedro Formation (note the painted rockfall hazard on the right). (**D**) Bryozoan fossils collected from La Vid Group. (**E**) Disused quarry exposing the Santa Lucía Formation. Source: prepared6 by the authors. Template of the 3D image: Google Earth.

Formigoso Formation (Figures 4 and 10B). Consists of black shales at the base with sandstones towards the middle and upper part. The basal shales contain fossils of benthic fossils (conulariids, bivalves, gastropods, trilobites, etc.) and nektonic fossils (cephalopods, hyolitids, and above all graptolites). Deposition began in a low-energy, oxygen-poor, shallow marine environment to less restricted conditions towards the top. It has been dated as Llandovery–Wenlock, but without further precision. It is worth noting that the numerous tectonic elements (faults, small thrusts, folds) in these rocks generate an anomalous thickness of the formation in this section that makes it difficult to study and teach.

San Pedro Formation (Figures 4 and 10C). Mainly made up of red sandstones, locally very ferruginous and with abundant volcanic pebbles. It contains brachiopod mould fossils, and some layers show a high degree of bioturbation. Its age is Middle Silurian–Basal Devonian.

La Vid Group. Made up of four formations with a wide range of lithologies: carbonate rocks (dolomites, encrinitic limestones, bioclastic limestones), siliceous rocks (shales) and mixed carbonate/siliceous rocks (marls). They are very rich in fossils of benthic organisms, especially brachiopods, corals, bryozoans (Figures 4 and 10D) and echinoderms. The formation has been dated as Early Devonian (Lochkovian–Emsian). In this section, the La Vid Group is not well developed and is partially covered by soil and debris.

Santa Lucía Formation. Comprises massive reefal limestones and is therefore very fossiliferous (Figures 4 and 10E). The main fossils are tabulate corals, rugose corals, stromatoporoids, brachiopods, bryozoans and echinoderms, among other groups. As it is a very compact and partially recrystallized limestone, the extraction of fossils without damaging them is very difficult. Its age ranges between Early and Middle Devonian (late Emsian–basal Eifelian).

Huergas Formation (Figures 4 and 10). Consists of shales, sandstones and sandy limestones, with black shales commonly containing nodule in horizons. Its age is Middle Devonian (basal Eifelian–Givetian). This formation usually occurs in depressed relief and is covered by soil or vegetation, so it is not usually visible.

Portilla Formation (Figures 4 and 10). Composed of reef limestones with numerous fossils of hermatypic organisms (mainly tabulate corals, rugose corals, bryozoans and brachiopods). Its age is Middle Devonian (Givetian). To access this formation, it is necessary to leave the road and go up the hill that leads to the axial zone of the Alba syncline.

### 4.1.3. Geosite 5. Middle Cambrian Trilobites and Echinoderms Site

The Oville Formation is a siliciclastic succession subdivided into three members [34,54] (that are easily recognisable throughout the Cantabrian Zone). The lower member, Genestosa Mb., reaches 70 m thick in the Luna Valley and is dominated by homogeneous green and grey shales, locally interrupted by up to five sandstone beds. According to Zamora et al. [55], this member contains the highest diversity of trilobites and echinoderms in the Cantabrian Zone (Figure 11). Some brachiopods, sponge spicules and several types of sclerites also occur in these beds [50]. All these fossils show a strong ochre colour (Figures 12 and 13) due to fossilisation by limonite replacement. Many of the trilobites come from moults, and it is usual to find echinoderm skeletons partially disarticulated, although some complete skeletons are also found.

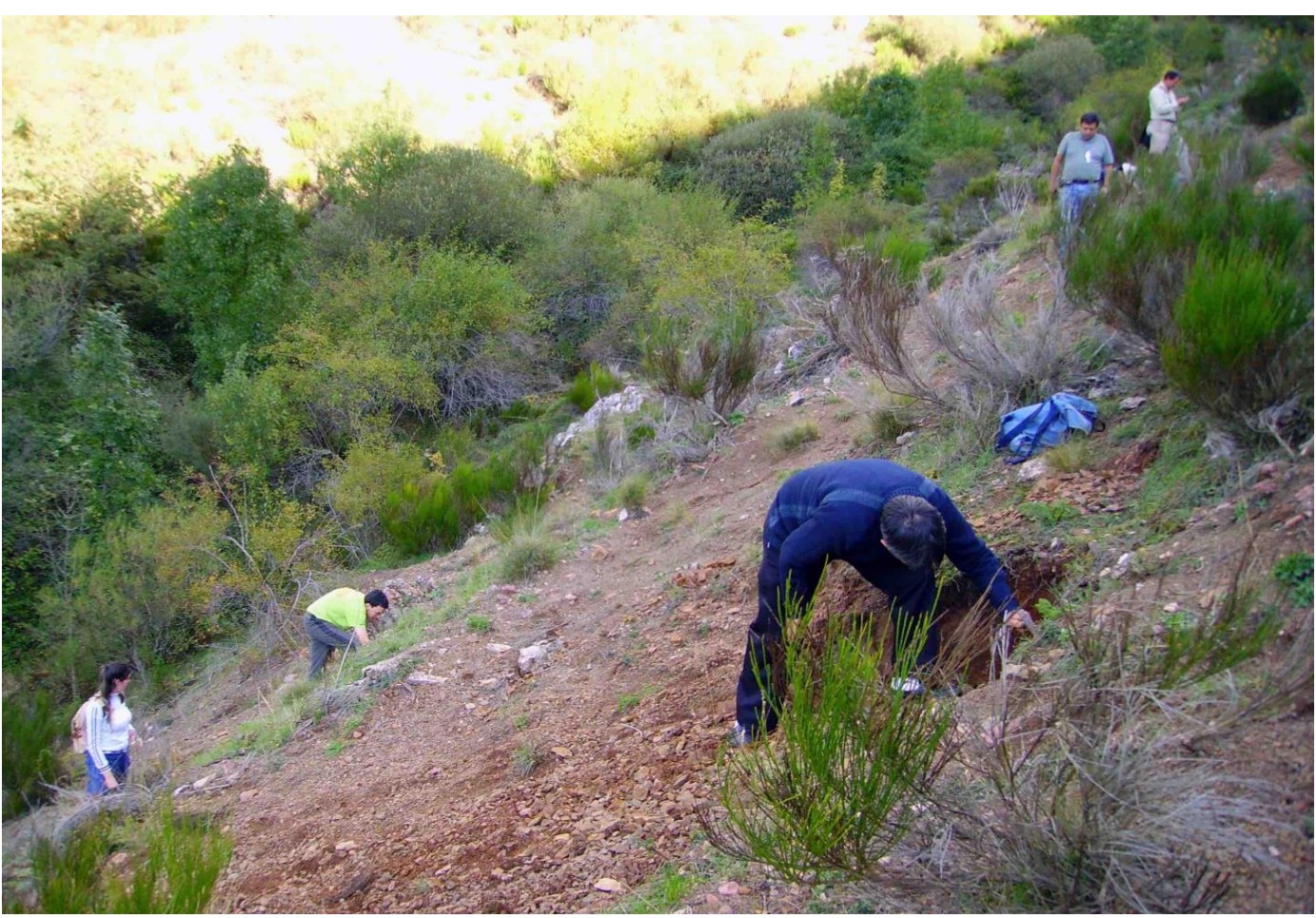

**Figure 11.** Field view of Geosite 5 during a visit by a group of palaeontologists. Note the absence of vegetation in this area due to the action of fossil collectors.

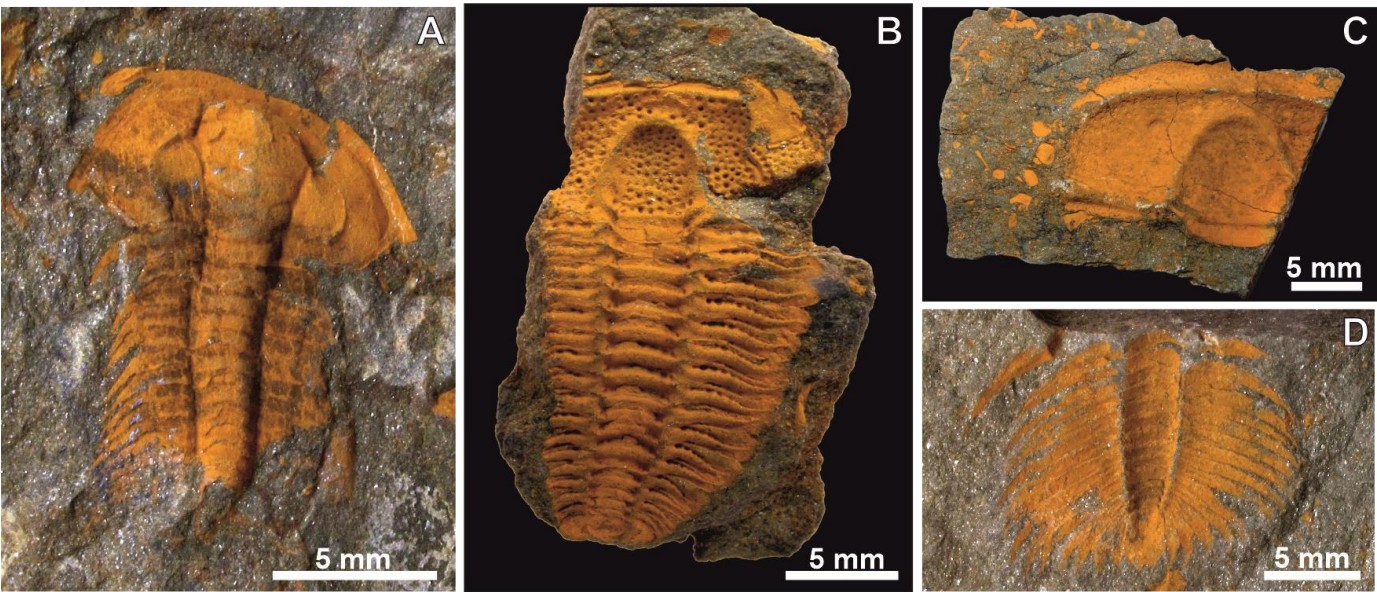

**Figure 12.** Cambrian trilobites from the Luna Valley (Geosite 5, Oville Formation). (**A**) *Eccaparadoxides* sp.; (**B**) *Solenopleuropsis* sp.; (**C**) *Conocoryphe* sp. (cephalon); (**D**) *Eccaparadoxides* sp. (pygidium). Note the yellowish ochre colour of these fossils due to the limonite replacement process. Source: prepared by the authors.

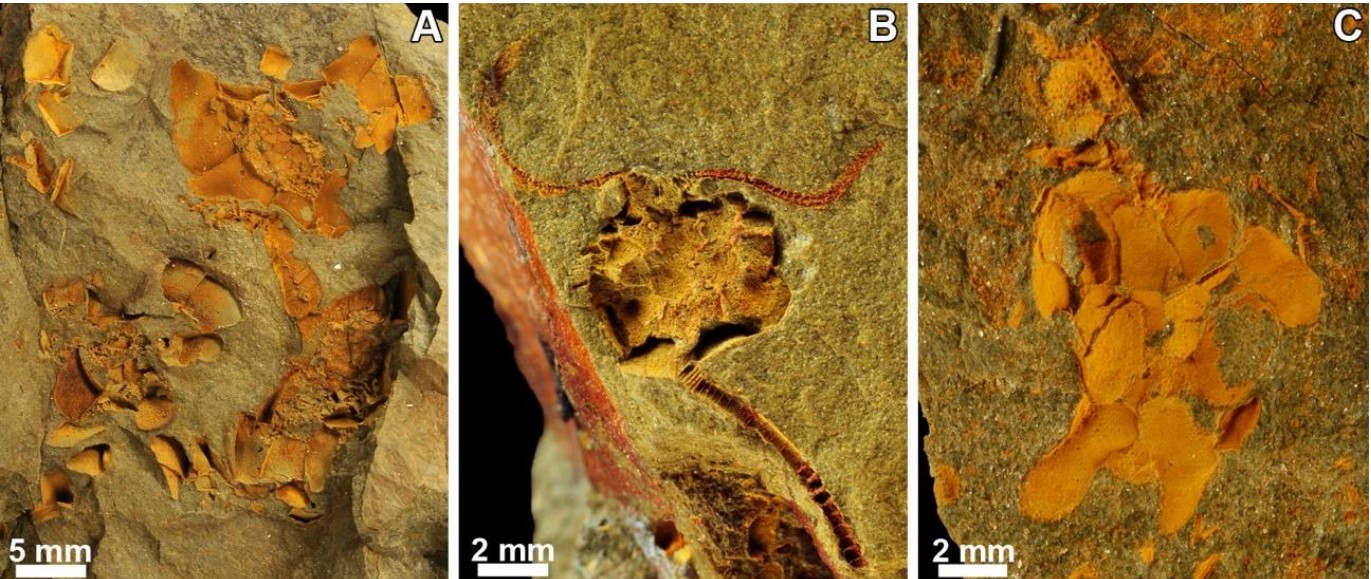

**Figure 13.** Cambrian echinoderms from the Luna Valley (Geosite 5, Oville Formation). (**A**) Three specimens of the cinctan *Lignanicystis barriosensis*. (**B**) The eocrinoid *Ubaghsicystis segurae*. (**C**). The armoured stylophoran *Ceratocystis* sp. Note the yellowish ochre colour of these fossils due to the limonitization process and the occurrence of disarticulated plates. Source: provided by Samuel Zamora ([55]).

The most fossil-rich layers of this palaeontological site are located on the eastern slope of the Luna River (Figures 2, 3, 5 and 11). Note that their exact location is not indicated in this work to avoid the uncontrolled collection of fossils.

The trilobite fauna (Figure 12) of this site have been studied since the 1960s. These studies have made it possible to determine the age of the Oville Formation (late Caesaraugustan–early Languedocian) and to carry out various taxonomic, palaeobiological and palaeogeographical studies [31,55].

Echinoderms (Figure 13) found to date include several taxa of cinctans [32,56], eocrinoids [57] and stylophoran [55]. The diachronic nature of the base of the Oville Formation allows for comparing these echinoderms with nearby fauna from similar environments but at different ages [55].

*4.2. Educational Activities and Materials*

Teaching activities began in the 1970s, when several researchers, in many cases from the University of Oviedo (Spain) and the University of Leiden (The Netherlands), began to work in the area not only for research but also to teach geology to undergraduate and graduate students.

Since 1999, the AEPECT (Asociación para la Enseñanza de las Ciencias de la Tierra, an association of secondary school teachers who teach Earth science subjects) has hosted courses and meetings, the last of which was held in April 2022. These courses generated several unpublished guides [58,59] that served as the background for field activities in the area, mostly aimed at secondary school students.

Students from several European universities have visited the outcrops in the Luna Valley, mainly the Lower Palaeozoic section (Geosite 3). Among the materials generated by this activity, the video by Fernández-Martínez and García del Canto [60], published by the University of León and available for viewing on the channel ULE.tv https://videos.unileon.es/video/5e95aa138f42088c638b4570 (accessed on 16 November 2022) stands out. In addition, a paper about the educational value of Los Barrios de Luna section that was published in 2007 [61] won the Ayala prize, awarded to the best paper about geological and mining heritage published in the Spanish magazine De Re Metallica.

Several scientific meetings have conducted field trips to these geosites; among them, we highlight the XXII Spanish Palaeontology Society (SEP), which issued a guide published by the University of León [50], and the international Palaeozoic Echinoderm Conference, whose work and field trip were published by Zamora and Rábano [55] (Figure 14).

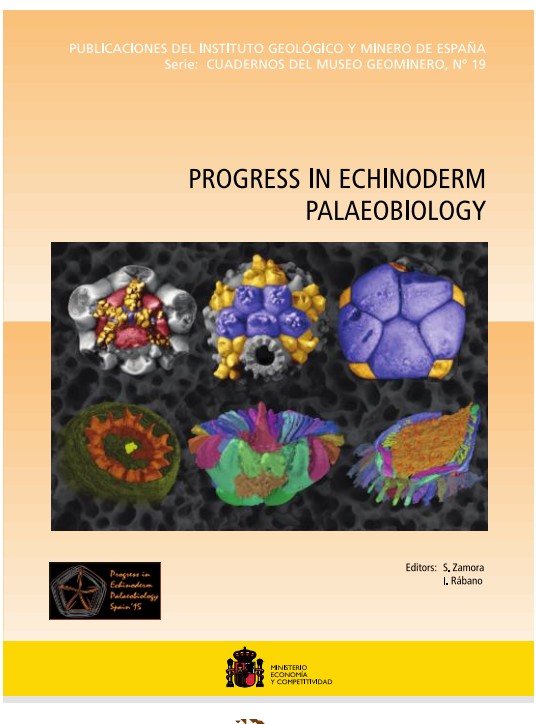

**Figure 14.** Cover page of the book containing information on the field trip during the Palaeozoic Echinoderm Conference (2015). Stop 14 of this field trip was in Geosite 5 of the Luna Valley complex geosite.

### 4.3. Outreach Activities and Materials

Outreach activities to the general public on Luna Valley's geological heritage began in 2000 and were developed by the Cuatro Valles Association (cuatrovalles.es) within the framework of several projects for the enhancement of the area's cultural and natural heritage.

In 2000, a trail called "Travelling to the Past" was designed [62] that covered the Lower Palaeozoic section (Geosite 3). To support this trail, a leaflet (Figure 15, right) and a flyer were printed and distributed in different places linked to nature tourism (rural tourism houses, restaurants). In 2005, an interpretation board was placed at the beginning of the trail (Figure 15, left). In addition, once a week during the summer months between 2001 and 2012, a guided tour was offered with an expert guide that garnered high appreciation among visitors.

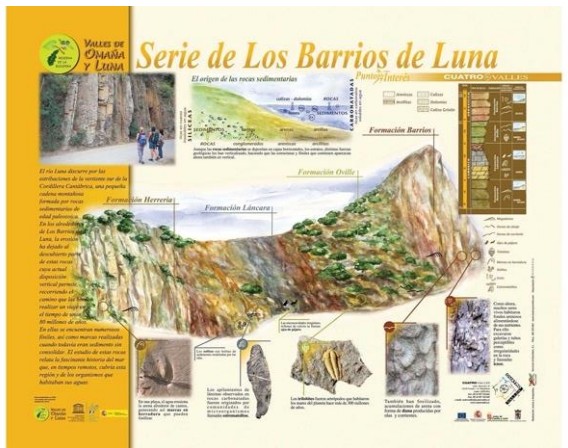 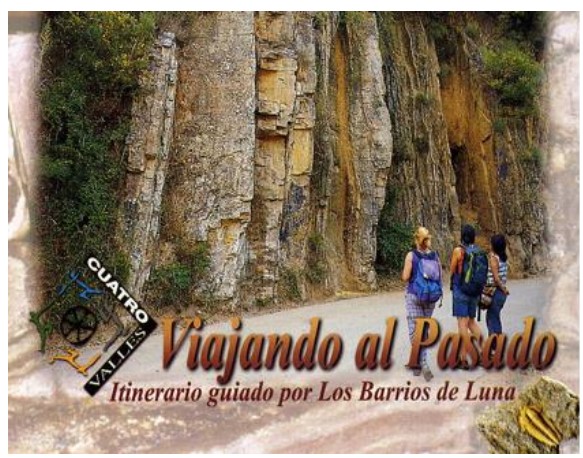

**Figure 15.** (**Left**). Interpretation board with information about the stratigraphy of the tour *Viajando al pasado* (Travelling back in time), placed at the beginning of the trail. (**Right**). Cover of the leaflet use for the promotion of the guided tour. Source: prepared by the authors.

In 2012–2013, the Cuatro Valles Association carried out an inventory of geosites in the area covered by this association. This inventory was published in a book [47] (Figure 16, left) that can be downloaded from the website (cuatrovalles.es). Out of 58 geosites, 28 were selected https://www.cuatrovalles.es/index.php/puntos-de-interes-geologico (accessed on 16 November 2022), and a small interpretation board was placed on each of these selected geosites (Figure 16, right). Each board includes a QR code that allows access to an app with detailed geological information, including the reconstruction of the process that created each geosite. Geosites 1, 2, 3 and 5 of this work are included in this guide, as well as 7 others single geosites located in the Luna Valley. Some of these sites also appear in more general materials focused on natural heritage, such as the Guide of the Natural Heritage of Cuatro Valles [63].

In 2013, promoted by the Town Council of Los Barrios de Luna and designed by Paleoymás https://www.paleoymas.com/dt_portfolios/centro-de-interpretacion-del-cambrico/ (accessed on 16 November 2022), the so-called Cambrian Interpretation Centre was created (Figure 17). Various specialists in the geology of the area participated in its design and contents [64]. This museum, located in the small town of Miñera de Luna (Figure 2), had restricted visits for several years. Today, it is a member of the Museos Vivos network https://museosvivos.com/ (accessed on 16 November 2022) and can be visited freely on request. As a complement to this interpretation centre, a CD was produced with a video and a brochure on the natural and cultural heritage of the municipality of Los Barrios de Luna.

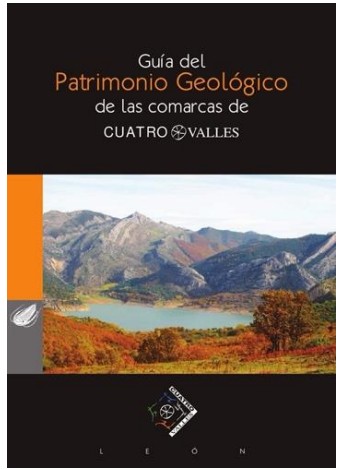
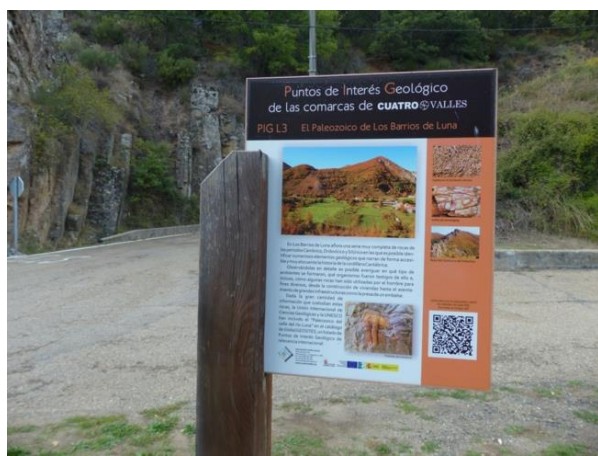

**Figure 16.** (**Left**). Cover of the guidebook on the geological heritage of the Cuatro Valles region. (**Right**). Small board located at the beginning of the trail that runs the Lower Palaeozoic section (Geosite 3). These panels are placed on the ground and have a QR code that gives access to geological information about the geosite.

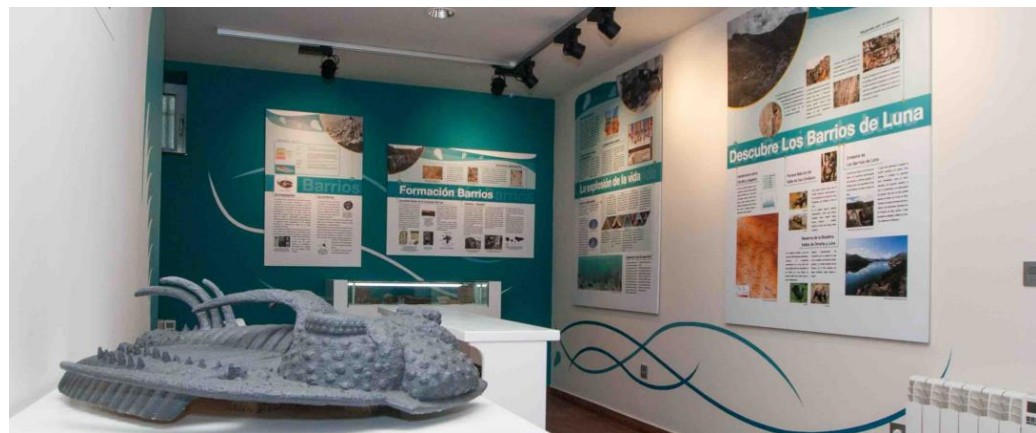

**Figure 17.** Cambrian Interpretation Centre in Miñera de Luna locality. This small museum explains the meaning of the explosion of life in the beginning of the Cambrian using fossils from the Luna Valley and other parts of our planet. Source: photo provided by Paleoymás.

The interpretative materials published by the Babia y Luna Natural Park do not contain much geological information and focus on glacial geomorphology. However, several leaflets mention the occurrence of fossils in the area, and the specific brochure on the municipality of Los Barrios de Luna refers to Geosites 3 and 5.

*4.4. Community Involvement*

Los Barrios de Luna is a very small municipality (308 inhabitants in the year 2021) and has the category of demographic desert (around 3 people per sq. km of land area). The average age of the population is 53.76 years; most people have no higher education, are retired, and live in the area only during the summer. The main economic base is livestock farming, producing local goods and tourism.

To date, the use and promotion of this region's geological heritage have essentially been carried out by (1) scientists researching in the area, (2) teachers who use these materials to teach, (3) the Cuatro Valles Association and (4) the town council of Los Barrios de Luna.

No studies have been conducted on the inhabitants' perceptions of natural or cultural heritage. Some testimonies collected during activities organised by the Valles de Omaña y Luna Biosphere Reserve indicate that knowledge of the importance of geological heritage in the area has been increasing since the creation of this Biosphere Reserve. However, the local population does not perceive the geosites as elements that can promote the socioeconomic development of the area.

This could explain why the local population has not been extensively involved in the local activities with the exception of some sessions to disseminate information about the area's geoheritage and aimed specifically at the local population. As a result of these experiences, many of the local inhabitants have been monitoring the outcrops, warning the guards when there is a group collecting fossils and asking people who walk the trail to take care of them.

The Spanish programme *Apadrina una roca* (Sponsor a Rock) has created a network of people who monitor the national geosites, reporting changes and warning of possible damages. This Luna Valley geosite has, to date, only four sponsors.

*4.5. Main Interest of the Complex and Single Geosites*

4.5.1. Scientific Interest

As indicated above, the complex geosite "Precambrian and Palaeozoic rocks of the Luna Valley" has been identified as a Global Geosite. The Global Geosite Project was promoted by the International Association for the Conservation of the Geological Heritage (ProGEO) and by the International Union of Geological Sciences (IUGS) in the late 1990s and now since 2020. Its main objective is to generate a list of sites of geological interest and global significance, i.e., geosites with a deep scientific interest.

The present scientific value of these outcrops has been described in numerous studies that have been published both in Ph.D. theses and in a great number of scientific papers (see list of recent contributions below). Several stratotypes (type sections) and parastratotypes of Neoproterozoic and Palaeozoic formations, such as Mora, Láncara, Barrios and La Serrona, are located in this area. Of particular interest is the fact that these sites are still the subjects of scientific studies, which has allowed us to improve our knowledge of the geological and biological history during the Palaeozoic.

As an example of recent contributions to this scientific knowledge, some works can be cited. Zamora and Smith [32] use some exquisitely preserved fossils the Middle Cambrian of Los Barrios de Luna for describing a new genus of *Cincta*, a stem group echinoderm. *Lignanicystis* nov. gen. (Figure 13A) has a strongly asymmetrical shape interpreted by the authors as an adaptation to life in higher water flow regimes.

Gutiérrez-Marco et al. [30] establish the record of the terminal Ordovician glaciation in the rocks of the Luna Valley and point out some stratigraphic and palaeontological features linked to this glaciation.

Gutiérrez-Alonso et al. [53] provide geological and geochronological data and arguments that support the idea that some ash-fall deposits found in several places in northern Iberia and related realms were produced by one super-eruption at a passive margin. One of the studied deposits was the tonstein located in Geosite 3 (Figure 8H).

The scientific study of this site has been made possible by the coexistence of several features. These include the good exposure of the rocks (due to the construction of roads mostly perpendicular to the orientation of beds), the verticality of the strata, the lack of large stratigraphic gaps and the occurrence of numerous sedimentary structures and both body fossils and ichnofossils.

Considering the five geosites described above, Geosite 3, which contains several stratotypes and has been studied in depth, and Geosite 5, bearing numerous and diverse fossils, are those of greatest scientific value (Table 1).

**Table 1.** Main interests of five geosites in the Luna Valley. The score and qualitative estimation for the complex geosite are from the Spanish Inventory of Geosites (IELIG). The qualitative scoring of single geosites has also been obtained using the IELIG inventory criteria [20].

| **Single and Complex Geosites** | **Scientific Value** | **Educational Value** | **Recreational Value** |
|---|---|---|---|
| 1. Angular unconformity at Irede de Luna | Low | Middle | No value |
| 2. Angular unconformity near Portilla de Luna | Middle | Middle | No value |
| 3. Lower Palaeozoic section at Los Barrios de Luna | High | High | Middle |
| 3. Lower-Middle Palaeozoic section at Los Barrios de Luna | Middle | Middle | Low |
| 5. Palaeontological site (trilobites and echinoderms) | High | Middle | No value |
| Complex geosite (IELIG) | Very high (7, 5) | High (7, 6) | High (7, 1) |

### 4.5.2. Educational and Recreational Interest

In-depth scientific knowledge of the area, accessibility, good exposure conditions and the availability of a large number of geological features are the factors that make this site of great educational value (Table 1).

This valley also has high natural and aesthetic value (Figure 18), as evidenced by its location within a natural park and a biosphere reserve (Figure 2). Thus, in addition to students and teachers, there is the occasional attendance of the general public interested in nature who usually walk trails or stop at the roadside viewpoints.

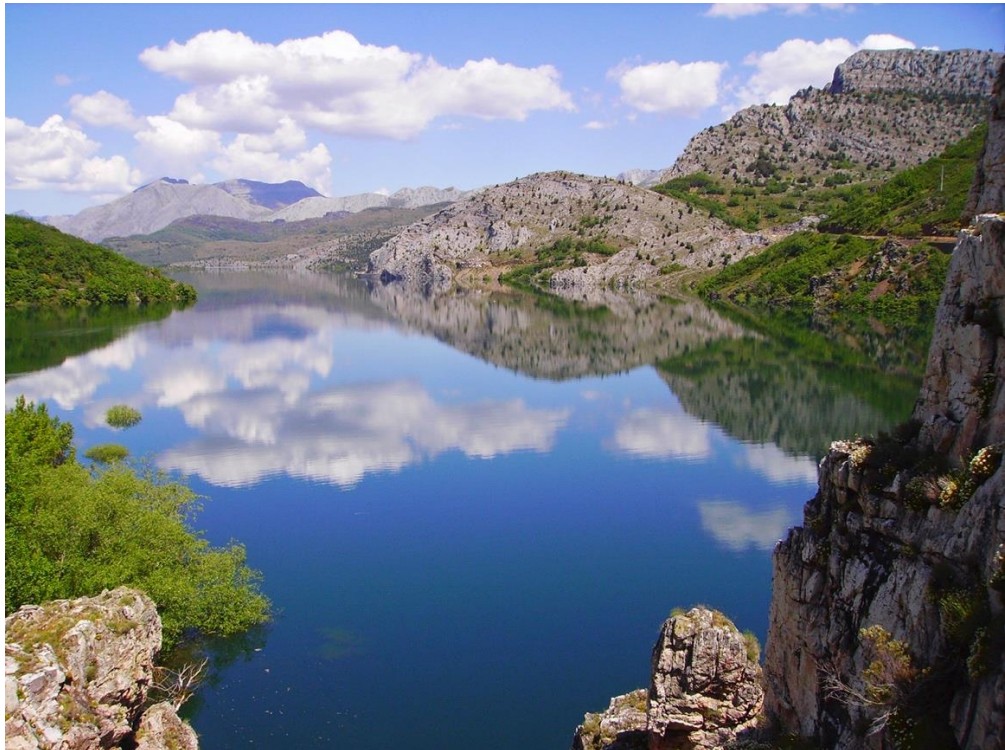

**Figure 18.** View from the closure of the Luna River reservoir, a point located in the final part of the lower Palaeozoic section (Geosite 3). It shows the aesthetic value of the area where this complex geosite is located.

Considering the five geosites, Geosite 1 (angular unconformity at the Irede de Luna locality) is of educational interest for secondary schools and universities because it allows observing this type of contact in the field and inferring the tectonic history of the two rock assemblages. However, Geosite 2 (the same geological feature at Portilla de Luna) is more accessible, it includes a large outcrop of the Mora Formation rocks and it allows us to analyse in detail the features of the rocks involved in this Neoproterozoic–Palaeozoic boundary. For this reason, Geosite 2 has important educational interest. To date, none of these two sites have been used for touristic or recreational purposes (Table 1).

In the case of the lower Palaeozoic section (Geosite 3), the educational and tourist value has been greatly enhanced by the development of various materials and activities. Indeed, this section is visited annually by students from multiple university majors related to Earth sciences, by postgraduate students, by teachers in training and by secondary school pupils. Geosite 4 (middle Palaeozoic section) has traditionally been used as a complementary section to Geosite 3 in teaching activities, but this use is now in decline.

Concerning Geosite 5, it has been used as an educational site for students of palaeontology-related subjects at university level for several decades. It shows how the fossils appear and how they can be collected and studied. Especially interesting is the fact that most of the trilobites do not correspond to trilobite corpses but to their moults (discarded carapaces). The occurrence of early echinoderms that are quite different from those that inhabit today's world is also interesting. The site has not been used for tourism or recreational purposes, although gathering fossils for commercial and collecting was a popular activity at the end of the 20th century.

*4.6. Conservation and Management Issues*

The elaboration of this section has deeply considered the geo-conservation considerations suggested by Prosser et al. [9] and Crofts et al. [1]. Additionally, and in order to summarise the degradation risk of the five single geosites, we have followed the model proposed by García-Ortiz et al. ([65]) (Table 2).

**Table 2.** Main heritage features of the five geosites in the Luna Valley.

| Single Geosites | Main Use | Problems | State of Conservation | Recommendations |
|---|---|---|---|---|
| 1. Angular unconformity at Irede de Luna | Educational | Access difficulties | Poor | Scientific use only |
| 2. Angular unconformity near Portilla de Luna | Educational | Face instability | Good | Improvement of the site (physical and material) |
| 3. Lower Palaeozoic section at Los Barrios de Luna | Scientific Educational Tourist | High degradation | Very poor | Take important actions for site safeguard and improve promotion |
| 3. Lower-Middle Palaeozoic section at Los Barrios de Luna | Scientific Educational | Risk of rockfall | Poor | Scientific use only |
| 5. Palaeontological site | Scientific Educational | Fossil plundering | Very poor | Scientific use only Virtual and ex-situ collections of fossils |

4.6.1. Geosite 1. Viewpoint to the Angular Unconformity at Irede de Luna

Conservation status. This site is the classic location for viewing the Neoproterozoic–Palaeozoic angular unconformity (Figure 6), and its use has been largely educational. However, in recent years, it has become less visited due to access problems. Because Geosite 1 is an extensive exposure site [9] its main threat is the degradation of exposure through weathering and vegetation encroachment. In addition, the abandonment of livestock farming has led to vegetation encroachment not only on the site itself but also along the access path to the viewpoint, making this access increasingly difficult.

This geosite also has a high degree of fragility related to the very high rate of erosion, which is due to the type of rocks (mostly shales and slates), the verticality of the slope and the lack of scrub around the angular unconformity. Thus, eroded materials have fallen on the Precambrian rocks. All these processes mean that the angle formed by the strata on either side of the unconformity (i.e., the fundamental geological feature) is not always visible.

Management issues. Includes periodic vegetation clearing of the path to the geosite in order to secure scientific access to the geological feature. For education goals, it would be important to build some kind of stable, vegetation-free area that would function as a viewpoint on the opposite slope. If these works were to be carried out, it would be advisable to signpost this access and provide geological interpretations of the site.

However, due to the disuse of this geosite and the need for the ongoing maintenance of the path, we propose abandoning this geosite for educational purposes and promoting Geosite 2 as an alternative.

### 4.6.2. Single Geosite 2. Point Showing the Angular Unconformity near Portilla de Luna

Conservation status. This outcrop is located at the side of a road, and in front of it there is a large place that can be used as a car park. Due to the ease of access and to the opportunity for viewing different geological features (Figure 7), it has been used extensively for teaching purposes in recent years.

The main risk of degradation is related to natural erosion, as the outcrop is situated on a vertical slope and the weather (with high temperature contrasts) causes the rocks to fall. To date, there is no evidence that the educational use of this site affects its conservation status.

Management issues. The use of Geosite 2 as a geoeducational alternative to the previous one reflects the realisation of promotional materials (boards, leaflets, virtual resources) linked to the site interpretation. Other necessary actions would include improving the car park and placing road signs indicating the potential presence of pedestrians. Some face stabilisation works may be necessary in the near future, but it is important to ensure that exposure is retained if the road is widened [9].

### 4.6.3. Single Geosite 3. Section of the Lower Palaeozoic Rocks at Los Barrios de Luna

As stated above, this classic section is of high scientific and educational interest. In addition, its touristic value has been reinforced with the creation of a trail and guided tour and with the creation of various outreach materials. It is easy to access, has a car park nearby and, as it is located on a road with very little traffic, is easy to walk around.

Conservation status. The road where this outcrop is located was built in the 1940s as a service road for the construction of the Luna River dam. For this reason, during the first years of the study, the section had optimal conditions for scientific works. At present, 70 years later, the section has a high degree of deterioration due to both natural and anthropic causes (Figures 19 and 20).

The natural vulnerability mainly affects the shales and sandstones. Part of the degradation is caused by breakage and rock fall due to weathering (Figure 19A), but this weathering is sometimes aggravated by human activities. For example, over the last few years, some shale layers rich in ichnofossils have collapsed, and in fossil-rich areas, some pits have been dug (Figure 19B). Recently, the presence of cattle farming in the upper part of some sectors has led to the fall of numerous limestone fragments. Another natural cause of deterioration is the concealment of the strata by the growth of vegetation and in many cases lichens. The latter, especially lichens of the *Rhizocarpon* group, are frequent on quartzite, the surface of which they can cover completely. When these lichens dry out, the rock takes on a black colour (Figure 19C) that prevents the direct viewing of its surface.

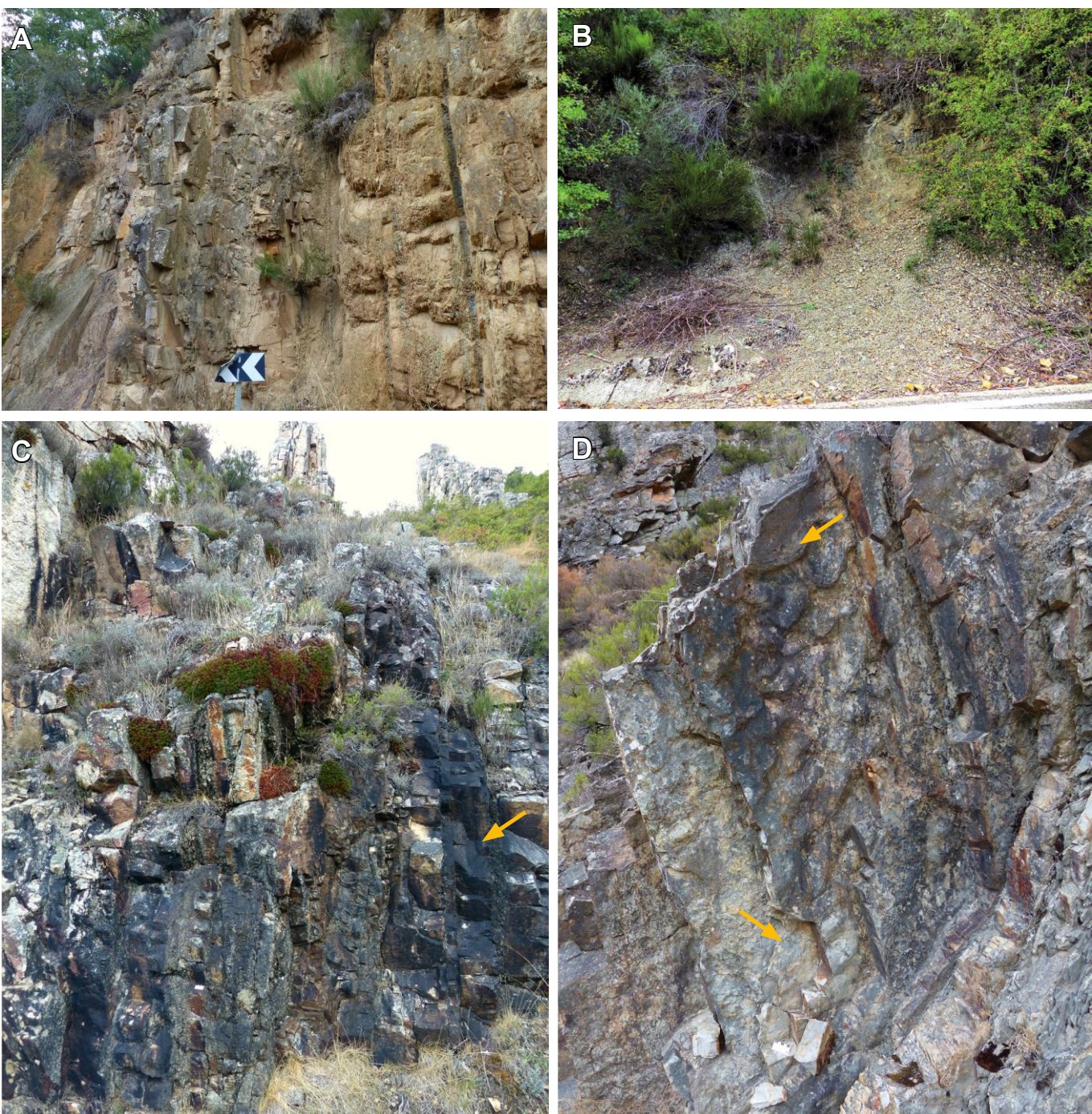

**Figure 19.** State of conservation of some of the geological features of Geosite 3. (**A**) Deteriorated sign due to rockfall. (**B**) Vegetation encroachment on the less resistant lithologies (shales) and the opening of a small pit made by fossil hunters. (**C**) Blackening of the surface of the strata by lichens and the deposit of manganese. (**D**) Current state of a surface with lichens (photo taken in 2022; compare with Figure 8F that shows this same exposure in 2009); it shows the blackening of the surfaces, the occurrence of marker traces and the intentional breakage in the lower part of the bed.

Another important cause of degradation has an anthropic origin (anthropogenic vulnerability) and is directly related to the geological interest of this section, i.e., with its use in scientific, tourist and, especially (because of the large number of people involved) educational activities. This section is mostly used for training purposes with university students. The numerous marks of geologists' hammers (Figures 19D and 20A–D) and the

indelible drawings made by teachers (Figure 20B,D) are frequent throughout the section. Another factor that is extremely deteriorating the value of this geosite is the extraction of rocks, sedimentary structures and fossils (Figures 19D and 20C,D), which are the key elements for the educational use of this section.

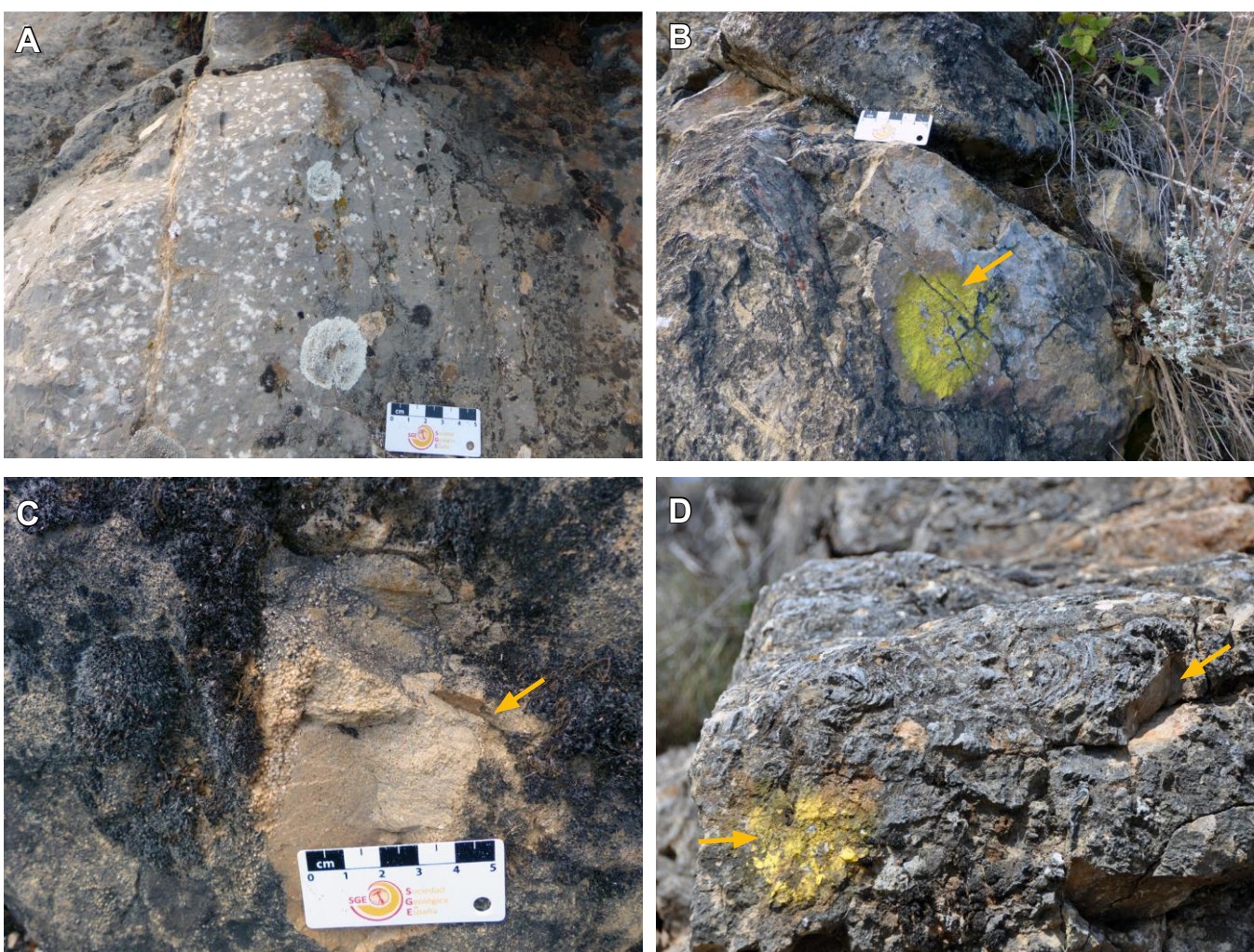

**Figure 20.** Signs of degradation at Geosite 3 caused by anthropic vulnerability. (**A**) Hammer impacts that leave white marks on the surface of a rock, making it impossible to detect sedimentary structures. (**B**) Drawings on the rock showing the location and pattern of some structures, in this case, microbialites. (**C**) Loss of a group of oncolites (see original feature in Figure 8D) by intentional hammering. (**D**) Broken (and currently missing) stromatolite whose location is marked by the yellow circle.

Management issues. Positive management of this site has to address three basic problems: high natural vulnerability, degradation due to anthropogenic vulnerability and the need to improve the current promotional actions. The first two problems must be tackled using the management instruments that are available in the natural areas where the complex geosite is located, specifically the Natural Park of Babia y Luna.

In this context, degradation caused by weathering (natural vulnerability) (Figure 19) can be partially eliminated by measures such as vegetation control, face stabilization, the removal of rock debris and slumped material and the cleaning of dead lichens and moss on the rocks. This work should be carried out by the Natural Park's maintenance team or by environmental volunteers organised by the park or by the Biosphere Reserve.

The anthropogenic degradation (Figures 19 and 20) is clearly due to the educational and recreational uses of this route. For this reason, the first step is to warn those institutions

that conduct geoeducational field work at this site about the delicate situation of the outcrop. This warning should include personalised letters to teaching staff, boards and signs requesting nonintervention on the rocks placed at the ends of the route and similar warnings also in teaching material such as leaflets.

In the meantime, and as a deterrence tool, the environmental guards in the area (which includes several different units) could be trained on the value of this site, which should be included in the list of sites to be monitored. A further step would be the inclusion of Geosite 3 in a special reserve area within the Babia y Luna National Park, although this would entail modifying the current regulations.

Despite its high degree of degradation, Geosite 3 performs very well as a geotrail for educational and recreational purposes. For this reason, we consider that the quality of both the physical site and the interpretation materials should be improved. In this sense, it would be interesting to review this geotrail and adapt it using the recommendations of Stolz and Mergele [66]. In addition, we consider particularly useful the development of tools such as augmented reality [67,68] and the QR-code-based virtual geotrails [69].

### 4.6.4. Single Geosite 4. Section of the Middle Palaeozoic Rocks at Los Barrios de Luna

This section has high scientific value. In the past, it was used for educational purposes, but at the moment this use is restricted. This is mainly due to its location, on a road with some traffic and low pedestrian visibility. It is also influenced by the fact that the road cuts through the strata obliquely, so that the exposure conditions are poor, and there are long distances to cover with little variation in geological features. Finally, there are dangerous areas due to rockfalls (Figure 10C).

Conservation status. Its main risk of degradation is due to natural vulnerability, principally because of the intense weathering processes that affect the rocks in this section. This weathering leads to a large amount of rock debris, slumped material, unstable faces and significant vegetation encroachment.

Management issues. In our opinion, this section should no longer be used for educational purposes because of the risk of physical harm to users and the poor exposure of the geological features.

### 4.6.5. Single Geosite 5. Palaeontological Site

As mentioned above, this site is of high scientific value. Since around the 1970s, it has also been used for training in palaeontology, especially for undergraduate students.

Conservation status. Due to the lithology and the type of fossilization, the site is very fragile. The most important degradation observed is due to trampling by horses grazing in the vicinity and, especially, by erosion linked to the gathering of fossils.

Spanish legislation is unclear on fossil collecting [70,71], but the long and indiscriminate gathering activity has contributed to the deterioration of the scientific value of the site. The progressive awareness of the geological heritage, the depletion of the site and the behaviour of the local population, who alert the environmental guards when they see groups on the site, have led to the area being abandoned for educational use.

Management issues. The poor state of conservation of this site and the progressive awareness of the scientific value of the fossils found there make it advisable to modify the use made of this site. Our proposals are twofold. One, maintain the scientific use of the site but require extraction permits from the management of Babia y Luna Natural Park. Two, transform the way the site is used for educational and recreational purposes. This involves making an interpreted exhibit of a small collection of fossils, for example in the Cambrian Interpretation Centre. This material would be at the service of interested teachers. A further step would be to digitize the collection and use this database as a way of promoting the geological heritage of the Luna Valley.

## 5. Discussion

In recent years, much progress has been made in geoconservation, especially in protected and conserved areas [1]. In this context, this work exposes a real example of the need to recognize the conservation status and threats affecting any geosite in use. In addition, the results of this study raise several questions about the use of geological heritage.

The five geosites analysed show several issues generated by the typical threats already identified in the literature. In this case, the main ones are the degradation of the geological exposures through weathering and lichen growth and vegetation encroachment. The solution to these problems involves vegetation control and periodic rock cleansing. As the complex geosite is located in a protected area (natural park), it is the responsibility of the administration of this natural park to carry out these actions.

Two of the five geosites analysed (3 and 5) have experienced threats from misuse, producing a very high deterioration. In other words, it is the geologists themselves who are altering the value of these sites. This suggests that geologists do not always consider the good practices that protected areas require of their visitors. Therefore, one of the actions to be considered in the near future is to seek ways of promoting awareness of the need to avoid damaging the sites we study. Thus, for these sites, our proposal is to carry out awareness actions but also for the environmental services to impose restrictions. In addition, any research or teaching activity involving sample collection in the area should be communicated to and registered in advance with the director of Babia y Luna Natural Park.

Despite the proposed restrictions, the scientific and educational interest of this complex geosite means that it should remain in use. To achieve this goal, it would be particularly useful to implement tools such as augmented reality or the QR-code-based virtual geotrails, which allows us to deepen our understanding of the geology of the sites without causing damage to them. Another very useful tool, especially for the conservation of geosite 5 (palaeontological site) is the museum, which is barely used. Hosting outreach activities at the museum aimed at specific audiences would allow not only teaching geological concepts but also educating on the importance of heritage.

The use of the indicated tools will increase the educational and training value of these geosites. However, it is also necessary to increase the recreational value for the general public who loves nature, especially the Valles de Omaña y Luna Biosphere Reserve and Babia y Luna Natural Park. These activities should involve the local population and focus on highlighting the problems of geological heritage and its value as a tool for social and cultural development.

However, before starting these activities, it would be advisable to carry out studies about community perception of the geosites and about the real potential of these sites to boost tourism.

## 6. Key Conclusions

An in-depth study of the heritage value, state of conservation and public use of a complex geosite has been carried out. The main conclusions derived from this study are the following:

1. Although most geosites are relatively robust, they are still subject to weathering that over time can generate a significant loss in value.
2. In general, it is assumed that educational activities do not cause damage to geological outcrops. However, the analysis carried out shows us that geosites can be used very badly. This misuse includes the looting or destruction of key geological features as well as some "well-intentioned vandalism," which, as in the case of exposed marker marks, should be avoided.

These two facts point to the need to monitor geosites, regardless of how they are used, and to include specific geoconservation actions in the management plans of protected areas.

3. The dissemination activities carried out to date have allowed the educational and recreational use of the geosites. However, this must be improved by introducing new technological tools such as augmented reality or virtual geotrails.

4. Especially in the case of geosites located within protected areas, the involvement of the local community in developing geological heritages is essential. In this sense, it is important to develop activities that capture the community's feelings about the geological heritage before developing any educational or land custody activities.

Finally, it is important to point out that these conclusions can be applied to many geosites located in protected areas where activities focused on flora and fauna take place but where the geological heritage has been neglected.

**Author Contributions:** Conceptualization: E.F.-M.; Investigation: E.F.-M., I.C., L.A. and R.C.; Resources: E.F.-M., L.A. and R.C.; Writing—Original Draft Preparation: E.F.-M. and L.A.; Writing—Review & Editing: E.F.-M. and I.C.; Supervision: E.F.-M. All authors have read and agreed to the published version of the manuscript.

**Funding:** This research received no external funding.

**Informed Consent Statement:** Not applicable.

**Data Availability Statement:** Not applicable.

**Acknowledgments:** The authors would like to thank Angel Gaspar, director of the Parque Natural de Babia y Luna, for his help in designing the management recommendations. To the people of Los Barrios de Luna and, in particular, to Raquel Familiar, for their help in monitoring the geosites and support in the activities carried out. Many thanks to Paleoymás and Samuel Zamora for the images they provided us with. Two of the authors (E.F.-M., I.C.V.) belong to the Research Group Q-GEO (University of León, Spain).

**Conflicts of Interest:** The authors declare no conflict of interest.

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
