# Peer review of "Factors in the Responsible Management of the Luna Valley Complex Geosite (NW Spain)—A Case Study"

_land, doi:10.3390/land11112082_

Round 1

Reviewer 1 Report

This is an interesting paper and I enjoyed reading it. However, there are essential weaknesses that need to be addressed.

1) The introductory/opening section should communicate a little clearer the literature gaps, as well as the study's aims & objectives in order to facilitate the flow of the study.

2) Overall there are good arguments and well researched points made in this paper, but I feel that author needs to take to a further level.  

I strongly recommend that you include the following references focused on the target journal and on the paper’s topics:

Ma, A., & Wu, Y. (2022). Total factor productivity of land urbanization under carbon emission constraints: a case study of Chengyu urban agglomeration in China. Economic Research-Ekonomska Istraživanja, 1-19.

Nedjah, N., de Macedo Mourelle, L., dos Santos, R. A., & dos Santos, L. T. B. (2022). Sustainable maintenance of power transformers using computational intelligence. Sustainable Technology and Entrepreneurship, 1(1), 100001. https://doi.org/10.1016/j.stae.2022.100001

Sapolaite, V., & Balezentis, T. (2022). The interplay of labour, land, intermediate consumption and output: a decomposition of the agricultural labour productivity for the Baltic States. Economic Research-Ekonomska Istraživanja, 35(1), 3512-3532.

3) The research is well-developed.

4) At the end of the ´Conclusion´ section, the author should include clear statements as to where research should now go.

5) Carefully check the references, so as to make sure they are all complete and follow the Guidelines to Authors.

6) Finally, when you submit the corrected version, please do check thoroughly, in order to avoid grammar, syntax or structure/presentation flaws.

Thank you for the opportunity to read the paper.

Author Response

The authors would like to thank you for your time and review. 

Reviewer 2 Report

The paper can be published after a minor check of English language.

Author Response

Thank you very much for taking the time to read the manuscript and for your nice comments, which ecourage us to continue our research in the field of geoheritage.

Two native English colleagues, who have made suggestions for improving the English and making it accessible to readers of all backgrounds, have read the new version of the manuscript.

Reviewer 3 Report

Numbering should be organized.

The review of literature for theoretical foundation is necessary.

In the introduction, the research goal should be stated clearly.

In the introduction, authors need to present why the place needs to be investigated.

In the method section, the procedure of this research needs to be descrbied logically. 

In the conclusion section, theoretical implications should be stated with logical manners. 

Author Response

Thank you very much for your time and review. Your suggestions have helped us a great deal, especially in improving the Introduction, Methods and Conclusions sections. 

Reviewer 4 Report

Dear Authors,

I want to thank for your submission and the opportunity to review an interesting manuscript focused on a analysis of the management of five key single geosites inside the complex geosite of Luna Valley located in the Northen-Western part of Spain.

Overall, the article is not well structured. I missed the methodology and discussion sections. Also, the topic is of relevance for the journal and the special issue.

Yet, as explained in the following detailed comments, I recommend several sugestions that I believe would help to improve the article. Therefore, I recommend that a major revision is needed. 

Introduction:

The authors begin the manuscript introducing the reader directly into the subject through a detailed presentation of the case study. The introduction should be focused on a brief presentation of the problem analyzed by the authors, highlighting the importance of geosites from scientific/educational and economic point of view, referring to relevant studies from the specialized literature. At the end of introduction you can mention the aim and objectives of the study. Moreover, you can refer to the significance of the study.

As a result of the fact that the presentation of the case study is extensive in the introduction, I recommend for the authors to create a distinct section called suggestively the description of the case study.

As mentioned above, the introduction is not clear enough and some concepts need to be better delimitated: eg geosites, their multiple importance from educational, cultural point of view, including their economic value (scientific tourism or other forms of tourism).

Methodology:

This section includes an inappropiate title: Interpretation methods and materials. As presented in this form by the authors it is more like the results section. So, the methodological section it should be renamed Material and Methods as it is mentioned in the template. You should focused on the methods used followed by their description and manner of application for the case study selected.

You should present several details regarding the evaluation of complex and single geosites in relation with the main interest of each. What method or criteria do you use? You mentioned at lines that the score and qualitative estimation for the complex geosite is after the Spanish Inventory of Geosites (IELIG). Is this estimation based on a several criteria?

Results

This section should be renumbered with 4.

The presentation of the geological characteristics (Geological setting of the study area) should be included in the results section. This can be followed by the others issues analysed: educational activities and materials, community involvement, etc.

You should add several ideas about community perception of geosites.

Discussion section

I missed a discussion section.

You should include in this section several information:

(1) the results should be better interpreted critically in order to reflect the relevance of the study. You should deeply analyze the assessment of the main interest of the single geosites and propose several solutions to increase the recreational value for the geosites that you judge to have a lower recreational value.

(2) you should make a better description of the contribution of the results emphasizing the importance of the study and what are the innovative aspects of your research;

 (3) You should mention the limitation of the study and future research. As a future research step, community’s participation in the promotion of geosites should be mentioned.

Minor comments:

You should mention the source of the figures 1 to 10, and 12. Within several figures you should mention the source of template of map (especially when you used aerial photographs) on which you added different data.

A renumbering of subsections and subsections is necessary. You have used the same numbering multiple times for different subsections.  

Author Response

Thank you very much for your time and review. Your suggestions have helped us a great deal, especially in improving the general outline of the MS and some of the main sections. 

Round 2

Reviewer 1 Report

-

Author Response

Both one of the authors and a native English colleague have  checked the English language. Thank you for this comment and for your valuable time.

Reviewer 4 Report

Dear Authors,

There is noticed an improvement of the manuscript, especially in terms of  its structure (presentation of methodology, discussion section).

Author Response

The organisation and structure of the manuscript have been improved especially thanks to your comments and suggestions. Thank you very much for your help.